# Using a Speed-Dependent Voigt Line Shape to Retrieve $O_2$ from Total Carbon Column Observing Network Solar Spectra to Improve Measurements of $XCO_2$

Authors: Joseph Mendonca[1], Kimberly Strong[1], Debra Wunch[1], Geoffrey C. Toon[2], David A. Long[3], Joseph T. Hodges[3], Vincent T. Sironneau[3], and Jonathan E. Franklin[4].

1. Department of Physics, University of Toronto, Toronto, ON, Canada

2. Jet Propulsion Laboratory, Pasadena, CA, USA

3. National Institute of Standards and Technology, Gaithersburg, MD, USA

4. Harvard John A. Paulson School of Engineering and Applied Sciences, Cambridge, MA, USA

*Correspondence to*: Joseph Mendonca (joseph.mendonca@utoronto.ca)

**Abstract**. High-resolution, laboratory, absorption spectra of the $a^1\Delta_g \leftarrow X^3\Sigma_g^-$ oxygen ($O_2$) band measured using cavity ring-down spectroscopy were fitted using the Voigt and speed-dependent Voigt line shapes. We found that the speed-dependent Voigt line shape was better able to model the measured absorption coefficients than the Voigt line shape. We used these line shape models to calculate absorption coefficients to retrieve atmospheric total columns abundances of $O_2$ from ground-based spectra from four Fourier transform spectrometers that are apart of the Total Carbon Column Observing Network (TCCON) Lower $O_2$ total columns were retrieved with the speed-dependent Voigt line shape, and the difference between the total columns retrieved using the Voigt and speed-dependent Voigt line shapes increased as a function of solar zenith angle. Previous work has shown that carbon dioxide ($CO_2$) total columns are better retrieved using a speed-dependent Voigt line shape with line mixing. The column-averaged dry-air mole fraction of $CO_2$ ($XCO_2$) was calculated using the ratio between the columns of $CO_2$ and $O_2$ retrieved (from the same spectra) with both line shapes from measurements made over a one-year period at the four sites. The inclusion of speed dependence in the $O_2$ retrievals significantly reduces the airmass dependence of $XCO_2$ and the bias between the TCCON measurements and calibrated integrated aircraft profile measurements was reduced from 1% to 0.4%. These results suggest that speed dependence should be included in the forward model when fitting near-infrared $CO_2$ and $O_2$ spectra to improve the accuracy of $XCO_2$ measurements.

## 1. Introduction

Accurate remote sensing of greenhouse gases (GHGs), such as $CO_2$, in Earth's atmosphere is important for studying the carbon cycle to better understand and predict climate change. The absorption of solar radiation by $O_2$ in the Earth's atmosphere is important because it can be used to study the properties of clouds and aerosols, and to determine vertical profiles of temperature and surface pressure. Wallace and Livingston (1990) were the first to retrieve total columns of $O_2$ from some of the discrete lines of the $a^1\Delta_g \leftarrow X^3\Sigma_g^-$ band of $O_2$ centered at 1.27 µm (which will be referred to bellow as the 1.27 µm band) using atmospheric solar absorption spectra from the Kitt

Peak observatory. Mlawer et al. (1998) recorded solar absorption spectra in the near-infrared (NIR) region to study
collision-induced absorption (CIA) in the $a^1\Delta_g \leftarrow X^3\Sigma_g^-$ band as well as two other $O_2$ bands. The spectra were
compared to a line-by-line radiative transfer model and the differences between the measured and calculated spectra
showed the need for better absorption coefficients in order to accurately model the 1.27 µm band (Mlawer et al.,
1998). Subsequently, spectroscopic parameters needed to calculate the absorption coefficients from discrete
transitions of the 1.27 µm band were measured in multiple studies (Cheah et al., 2000; Newman et al., 2000, 1999;
Smith and Newnham, 2000), as was collision-induced absorption (CIA) (Maté et al., 1999; Smith and Newnham,
2000), while  Smith et al. (2001) validated the work done in Smith and Newnham (2000) using solar absorption
spectra.
The 1.27 µm band  is of particular importance to the Total Carbon Column Observing Network (TCCON) (Wunch
et al., 2011). TCCON is a ground-based remote sensing network that makes accurate and precise measurements of
GHGs for satellite validation and carbon cycle studies. Using the $O_2$ column retrieved from solar absorption spectra,
the column-averaged dry-air mole fraction of $CO_2$ (XCO2) has been shown to provide better precision than using the
surface pressure to calculate XCO2 (Yang et al., 2002). The $O_2$ column is retrieved from the 1.27 µm band because
of its close proximity to the spectral lines used to retrieve $CO_2$,  thereby reducing the impact of solar tracker mis-
pointing and an imperfect instrument line shape (ILS) (Washenfelder et al., 2006). To improve the retrievals of $O_2$
from the 1.27 µm band, Washenfelder et al. (2006) found that adjusting the spectroscopic parameters in HITRAN
2004 (Rothman et al., 2005) decreased the airmass and temperature dependence of the $O_2$ column. These revised
spectroscopic parameters were included in HITRAN 2008 (Rothman et al., 2009). Atmospheric solar absorption
measurements from this band made at the Park Falls TCCON site by  Washenfelder et al. (2006) were the first
measurements to observe the electric-quadrupole transitions (Gordon et al., 2010).  Leshchishina et al. (2011, 2010)
subsequently used cavity-ring-down spectra to retrieve spectroscopic parameters for the 1.27 µm band using a Voigt
spectral line shape and these parameters were included in HITRAN 2012 (Rothman et al., 2013). Spectroscopic
parameters for the discrete spectral lines of the $O_2$ 1.27 µm band from HITRAN 2016 (Gordon et al., 2017) are very
similar to HITRAN 2012 except that HITRAN2016 includes improved line positions reported by Yu et al. (2014).
Extensive spectral line shape studies have been performed for the $O_2$ A-band, which is centered at 762 nm and used
by the Greenhouse Gases Observing Satellite (GOSAT) (Yokota et al., 2009) and the Orbiting Carbon Observatory-
2 (OCO-2) satellite (Crisp et al., 2004) to determine surface pressure. Studies have shown that the Voigt line shape
is inadequate to describe the spectral line shape of the discrete $O_2$ lines in the A-band. Dicke narrowing occurs when
the motion of the molecule is diffusive due to collisions changing the velocity and direction of the molecule during
the time that it is excited. This diffusive motion is taken into account by averaging over many different Doppler
states resulting in a line width that is narrower than the Doppler width (Dicke, 1953). Long et al. (2010) and Predoi-
Cross et al. (2008) found it necessary to use a spectral Line shape model that accounted for Dicke narrowing when
fitting the discrete lines of the $O_2$ A-band. Line mixing, which occurs when collisions transfer intensity from one
part of the spectral band to another (Lévy et al., 1992), was shown to be prevalent in multiple studies (Predoi-Cross
et al., 2008; Tran et al., 2006; Tran and Hartmann, 2008). Tran and Hartmann (2008) showed that including line
mixing when calculating the $O_2$ A-band absorption coefficients reduced the airmass dependence of the $O_2$ column
retrieved from TCCON spectra. When fitting cavity ring-down spectra of the $O_2$ A-band, Drouin et al. (2017) found
it necessary to use a speed-dependence Voigt line shape, which takes into account different speeds at the time of
collision (Shannon et al., 1986), with line mixing to properly fit the discrete spectral lines of the $O_2$ A-band.
The need to include non-Voigt effects when calculating absorption coefficients for the $O_2$ 1.27 µm band was first
shown in Hartmann et al. (2013) and Lamouroux et al. (2014). In  Hartmann et al. (2013) and Lamouroux et al.
(2014), Lorentzian widths were calculated using the re-quantized classical molecular-dynamics simulations
(rCMDSs) and used to fit cavity-ring-down spectra with a Voigt line shape for some isolated transitions in the $O_2$
1.27 µm band. The studies concluded that a Voigt line shape is insufficient for modeling the spectral lines of the $O_2$
1.27 µm band and that effects such as speed dependence and Dicke narrowing should be included in the line shape
calculation.
In this study, air-broadened laboratory cavity-ring-down spectra of the $O_2$ 1.27 µm band were fitted using a spectral
line shape that takes into account speed dependence. The derived spectroscopic parameters for the speed-dependent
Voigt line shape were used to calculate absorption coefficients when fitting high-resolution solar absorption spectra.
Using these new $O_2$ total columns, and the simultaneously measured $CO_2$ total columns, using the updated line
shape model described by Mendonca et al. (2016), to calculate $XCO_2$ and compared these results with $XCO_2$
retrieved using a Voigt line shape. Section 2 details the formulas used to calculate absorption coefficients using
different spectral line shapes. In Section 3, we describe the retrieval of spectroscopic parameters from three air-
broadened cavity-ring-down spectra fitted with a speed-dependent Voigt line shape. For Section 4, the speed-
dependent line shape along with the retrieved spectroscopic parameters is used to fit solar absorption spectra from
four TCCON sites and retrieve total columns of $O_2$, which is compared to $O_2$ retrieved using a Voigt line shape. In
Section 5, we investigate the change in the airmass dependence of $XCO_2$ with the new $O_2$ retrievals. In Section 6, we
discuss our results and their implications for remote sensing of greenhouse gases.
**2. Absorption Coefficient Calculations**
**2.1 Voigt Line Shape**
The Voigt line shape is the convolution of the Lorentz and the Gaussian profiles, which model pressure and Doppler
broadening of the spectral line respectively. The corresponding absorption coefficient, $k$, at a given wavenumber $v$
becomes :

$$k(v) = N \sum_j S_j \left(\frac{1}{\gamma_{D_j}}\right) \left(\frac{\ln(2)}{\pi}\right)^{1/2} \left(Re[c(v, x_j, y_j)]\right) \tag{1}$$

where $N$ is the number density, $S_j$ is the line intensity of spectral line $j$, $\gamma_{D_j}$ is the Doppler half-width (HWHM), c is
the complex error function, and

$$x_j = \frac{(v-v_j^o-P\delta_j^o)}{\gamma_{D_j}} (\ln(2))^{1/2}, \quad y_j = \frac{\gamma_{L_j}}{\gamma_{D_j}} (\ln(2))^{1/2} . \tag{2}$$

Here, $v_j^o$ is the position of the spectral line $j$, $P$ is the pressure, and $\delta_j^o$ is the pressure-shift coefficient. The Lorentz
half-width, $\gamma_{L_j}$, is calculated using:

$$\gamma_{L_j}(T) = P \, \gamma_{L_j}^o \left(\frac{296}{T}\right)^n \tag{3}$$

where $\gamma_{L_j}^o$ is the air-broadened Lorentz half-width coefficient (at reference temperature 296 K) and $n$ is the exponent
of temperature dependence. The Voigt line shape assumes that pressure broadening is accurately represented by a
Lorentz profile calculated for the stastical average velocity at the time of collission.

### 2.2 Speed-Dependent Voigt Line Shape

The speed-dependent Voigt line shape refines the pressure broadening component of the Voigt by calculating
multiple Lorentz profiles for different speeds at the time of collision. The final contribution from pressure
broadening to the speed-dependent Voigt is the weighted sum of Lorentz profiles (weighted by the Maxwell-
Boltzmann speed-distribution) calculated for different speeds at the time of collision. The speed-dependent Voigt
line shape (Ciuryło, 1998) with the quadratic representation of the Lorentz width and pressure shift (Rohart et al.,
1994) is:

$$k(v) = N \left(\frac{2}{\pi^{\frac{3}{2}}}\right) \sum_j S_j \int_{-\infty}^{\infty} e^{-V^2} V \left( \tan^{-1} \left[ \frac{x_j - Ba_{\delta_j}(V^2 - 1.5) + V}{y_j(1 + a_{\gamma_{L_j}}(V^2 - 1.5))} \right] \right) dV \tag{4}$$

where $a_{\gamma_{L_j}}$ is the speed-dependent Lorentz width parameter (unitless) for line $j$, $a_{\delta_j}$ is the speed- dependent pressure-
shift parameter (unitless), $B$ is $\frac{(\ln(2))^{1/2}}{\gamma_{D_j}}$, V is the ratio of the absorbing molecule's speed to the most probable speed
of the absorbing molecule, and all other variables are defined before.

### 3. Fitting Laboratory Spectra

$O_2$, unlike $CO_2$ and $CH_4$, cannot produce an electric dipole moment and therefore should not be infrared active.
However, $O_2$ has two unpaired electrons in the ground state that produce a magnetic dipole moment. Due to the
unpaired electrons in the ground state $(X^3\Sigma_g^-)$ the rotational state $(N)$ is split into three components which are given
by $J = N-1$, $J = N$, and $J = N+1$, while in the upper state $(a^1\Delta_g)$, $J = N$. When labeling a transition, the following
nomenclature is used $\Delta N(N'')\Delta J(J'')$ (Leshchishina et al., 2010), where $\Delta N$ is the difference between $N'$ in the upper
state and $N''$ in the lower state, $\Delta J$ is the difference between $J'$ in the upper state and $J''$ in the lower state. The
magnetic transitions of $a^1\Delta_g \leftarrow X^3\Sigma_g^-$ allow for $\Delta J=0, \pm1$. This leads to 9 branches observed: $P(N'')Q(J'')$,
$R(N'')Q(J'')$, and $Q(N'')Q(J'')$, for $\Delta J=0$, $O(N'')P(J'')$, $P(N'')P(J'')$, and $Q(N'')P(J'')$, for $\Delta J=-1$, and $S(N'')R(J'')$,
$R(N'')R(J'')$, and $Q(N'')R(J'')$, for $\Delta J=1$.
Absorption coefficients for three room temperature air-broadened (NIST Standard reference materal® 2659a
containing 79.28 % $N_2$, 20.720(43) % $O_2$, 0.0029 % Ar, 0.00015 % $H_2O$, and 0.001 % other compounds) spectra
were measured at the National Institute of Standards and Technology (NIST) using the frequency-stabilized cavity-
ring-down spectroscopy (FS-CRDS) ) technique (Hodges, 2005; Hodges et al., 2004). The absorption spectra were
acquired at pressures of 131 kPa, 99.3 kPa, and 66.9 kPa, at temperatures of 296.28 K, 296.34 K, and 296.30 K
respectively. Figure 1a shows the three measured absorption spectra.  A more detailed discussion of the present FS-
CRDS spectrometer can be found in Lin et al. (2015).
The spectra were fitted individually using a Voigt line shape (Eq. 1), with $S_j$, $\gamma_{L_j}^o$, and $\delta_j^o$ for the main isotope of the
magnetic dipole lines of the $O_2$ 1.27 µm band for lines with an intensity greater than $7.0 \times 10^{-28}$ $cm^{-1}$/(molecule $cm^{-2}$).
The spectroscopic parameters measured in Leshchishina et al. (2011) for the spectral lines of interest were used as
the a priori for the retrieved spectroscopic parameters. The  line positions were left fixed to the values measured in
Leshchishina et al. (2011), and all other $O_2$ spectral lines (intensity less $7.0 \times 10^{-28}$ $cm^{-1}$/(molecule $cm^{-2}$)) were
calculated using a Voigt line shape with spectroscopic parameters from HITRAN 2012 (Rothman et al., 2013).
Spectral fits were done using the lsqnonlin function in Matlab, with a user-defined Jacobian matrix. The Jacobian
was constructed by taking the derivative of the absorption coefficients with respect to the parameters of interest.
Using an analytical Jacobian instead of the finite difference method is both computationally faster and more
accurate. The Voigt line shape was calculated using the Matlab code created by Abrarov and Quine (2011) to
calculate the complex error function and its derivatives. To take collision-induced absorption (CIA) into account, a
set of 50 Legendre polynomials were added together by retrieving the weighting coefficients needed to add the
polynomials to fit the CIA for each spectrum. Figure 1b shows the residual (measured minus calculated absorption
coefficients) when using a Voigt line shape with the retrieved spectroscopic parameters. The plot shows that residual
structure still remains for all three spectra. The Root Mean Square (RMS) residual values for the spectra are given
by the legend at the side of the plot.
Figure 2 is the same plot as Figure 1 but for the P(11)P(11), P(11)Q(10), P(9)P(9), and P(9)Q(8) spectral lines only.
Figure 2b shows that for all four spectral lines there is a "W" shaped residual at the line center. The P(11)P(11) line
was also measured by  Hartmann et al. (2013) at pressures ranging from 6.7 to 107 kPa. Figure 5 of  Hartmann et al.
(2013) shows the P(11)P(11) line at a pressure of 66.7 kPa, which is approximately the pressure of the 66.9 kPa
spectrum (blue spectrum in Figure 1 and 2). When one compares the blue residual of the P(11)P(11) line in Figure
2b to that of the residual of the left panel of Figure 5 of  Hartmann et al. (2013), one can see that the residuals are the
same. Figure 6 of  Hartmann et al. (2013) show that the amplitude of the residual increases with decreasing pressure,
which is also seen in Figure 2b. Figure 3 of  Lamouroux et al. (2014) shows the same "W" residual for the P(9)P(9)
lines and that the amplitude of the residual increases with decreasing pressure (although for lower pressures)
consistent with the results for the P(9)P(9) line in Figure 2b.
Figure 1c shows the residual when using the speed-dependent Voigt (Eq. 4) to fit each spectrum individually. To use
Eq. (4) requires integration over all possible speeds, which is not computationally practical, so we employ the
simple numerical integration scheme as was done by Wehr (2005). When fitting the spectra, parameters $S_j$, $\gamma_{L_j}^o$, $\delta_j^o$,
$a_{\gamma_{L_i}}$ and $a_{\delta_j}$ were retrieved for lines of intensity greater than $7.0 \times 10^{-28}$ cm$^{-1}$/(molecule cm$^{-2}$), while all other $O_2$ lines
were calculated using a Voigt line shape and spectroscopic parameters from HITRAN 2012 (Rothman et al., 2013b).
The Jacobian matrix was created by taking the derivative with respect to each parameter of interest, as was done
with the Voigt fits. By taking speed-dependent effects into account, the residuals were reduced to 25 times smaller
than those for the Voigt fit and the RMS residuals (given in the legend of Figure 1c) are 10 times smaller. However,
some residual structure still remains, which is more evident in the in the Q and R branches than the P branch. Figure
2c shows the four lines in the P branch, as discussed when analyzing the Voigt fits. A small residual "W" remains at
line center, as well as residuals from weak $O_2$ lines.
Figure 3 shows the averaged intensity, Lorentz width coefficient, pressure shift coefficient, and speed-dependent
shift coefficient of the 1.27 μm $O_2$ band, retrieved from the three spectra, plotted as a function of quantum number
m which is m=-J (where J is the lower state rotational quantum number) for the P-branch lines, m=J for the Q-
branch lines, and m=J+1 for the R-branch lines. The intensity, Lorentz widths, and pressure shifts show a m
dependence for these parameters for the P and R sub-branches. The measured Lorentz widths and pressure shifts for
the Q sub-branches show a m dependence but are not as strong as the P and R sub-branches. This is because the Q
branch lines are broadened enough to blend with each other since they are spaced closer together than the P or R
branch lines. Figure 1c shows that some of the residual structure in the Q branch increases with pressure and is
partly due to the blending of these transitions as the pressure increases. The weak $O_2$ absorption lines also blend in
with the Q branch, contributing to the residual structure in Figure 1c. We tried retrieving the spectroscopic
parameters for the weak $O_2$ absorption lines, but since they were overlapping with the strong $O_2$ lines, it was not
possible. Figure 4a shows the retrieved speed-dependent width parameter averaged over the three spectra, plotted as
a function of m, showing that it increases with m. Error bars correspond to the 2σ standard deviation and are large
regardless of sub-branch. Figure 4b shows the retrieved speed-dependent width for the PQ sub-branch for the
different pressures. The speed-dependent width shows the same m dependence regardless of pressure, but also
increases with decreasing pressure as is the case for sub-branches. It should be noted that the speed-dependent width
parameter should be independent of pressure.
**4. Fitting Solar Spectra**
High-resolution solar absorption spectra were measured at four TCCON sites using a Bruker IFS 125HR FTIR
spectrometer with a room temperature InGaAs detector at a spectral resolution of 0.02 cm$^{-1}$ (45 cm maximum
optical path difference). The raw interferograms recorded by the instrument were processed into spectra using the
I2S software package (Wunch, D. et al., 2015) that corrects solar intensity variations (Keppel-Aleks et al., 2007),
phase errors (Mertz, 1967), and laser sampling errors (Wunch, D. et al., 2015), and then preforms a fast Fourier
transform to convert the interferograms into spectra (Bergland, 1969). The GGG software package (Wunch, D. et
al., 2015) is used to retrieve total columns of atmospheric trace gases. GFIT is the main code that contains the
forward model, which calculates a solar absorption spectrum using a line-by-line radiative transfer model and an
iterative non-linear least square fitting algorithm that scales an a priori gas profile to obtain the best fit to the
measured spectrum. A priori profiles for GHGs are created by an empirical model in GGG that is based on
measurements from the balloon-borne JPL MkIV Fourier Transform Spectrometer (FTS) (Toon, 1991), the
Atmospheric Chemistry Experiment (ACE) FTS instrument aboard SCISAT (Bernath et al., 2005), and in situ
GLOBALVIEW data (Wunch et al., 2011). Temperature and pressure profiles, as well as $H_2O$ a priori profiles are
generated from the National Centers for Environmental Prediction (NCEP) data. The calculations are performed for
71 atmospheric layers (0 km to 70 km), so all a priori profiles are generated on a vertical grid of 1 km.
In the current GGG software package (Wunch, D. et al., 2015), the forward model of GFIT calculates absorption
coefficients for the discrete lines of the $O_2$ 1.27 µm band using a Voigt line shape and spectroscopic parameters
from  Washenfelder et al. (2006a) and  Gordon et al. (2010). To take CIA into account, absorption coefficients are
calculated using a Voigt line shape and spectroscopic parameters from the foreign-collision-induced absorption
(FCIA) and self-collision-induced absorption (SCIA) spectral line lists provided with the GGG software package
(Wunch, D. et al., 2015). Spectroscopic parameters in the FCIA and SCIA line lists were retrieved by Geoff Toon by
fitting the laboratory spectra of  Smith and Newnham (2000). This was done by retrieving the integrated absorption
at every 1 cm$^{-1}$ of the spectrum and using a Voigt line shape, with fixed Lorentz width and no pressure shift. In
GFIT, a volume scale factor is retrieved for the CIA and discrete lines separately so that the $O_2$ column is derived
from the discrete lines of the 1.27 µm band only. Airglow is not considered when fitting the 1.27 µm band since the
spectrometer views the sun directly, and airglow is overwhelmed by such a bright source. The continuum level and
tilt of the 100% transmission level is fitted using a weighted combination of the first two Legendre polynomials.
Absorption coefficient for all other trace gases are calculated using a Voigt line shape and spectroscopic parameters
from the atm.101 line list (Toon, G. C., 2014a) and solar lines are fitted using the solar line list (Toon, G. C.,
2014b).
Figure 5 shows the spectral fit to a solar absorption spectrum recorded at Eureka on March 27, 2015, at a solar
zenith angle (SZA) of 81.32$^{o}$ (airmass of 6.3). This spectrum is an average of 5 Eureka scans. The TCCON standard
is single scan but 5 scans were averaged to decrease the noise. The measured spectrum (red circles), calculated
spectrum (black circles) and transitions from all gases in the window (colored lines, refer to the legend for different
gases) are shown in Figure 5b. The residual obtained using a Voigt line shape to calculate the discrete lines of the $O_2$
1.27 µm band is shown in red in Figure 5a. The blue residual is the result of using a speed-dependent Voigt line
shape with the spectroscopic parameters retrieved from fitting the absorption coefficients in Section 3. To decrease
the amount of time it takes to calculate the absorption coefficients, the quadratic-Speed Dependent Voigt (qSDV)
computational approach of Ngo et al. (2013) and Tran et al. (2013)was used instead of Eq. (4) since it requires the
Voigt calculation only twice, while Eq. (4) requires numerical integration scheme with 33 iterations. The
temperature-dependent parameter of the Lorentz width of the discrete lines of the $O_2$ 1.27 µm band reported in
HITRAN 2012 was used to take temperature dependence into account for $\gamma_{L_j}(T)$. There was only a slight
improvement in the fit residuals with the new absorption coefficients (using the qSDV), as seen in Figure 5a.
Absorption coefficients calculated with the qSDV were used to retrieve total columns of $O_2$ from solar spectra
recorded over a one year period at TCCON sites in Eureka (eu) (Nunavut, Canada) (Batchelor et al., 2009; Strong et
al., 2017), Park Falls (pa) (Wisconsin, U.S.A) (Washenfelder et al., 2006; Wennberg et al., 2017) , Lamont (oc)
(Oklahoma, U.S.A) (Wennberg et al., 2017b), and Darwin (db) (Australia) (Deutscher et al., 2010; Griffith et al.,
2017). In total 131 124 spectra were fitted using the qSDV and the average root mean square (RMS) residual of the
fit only decreased by 0.5 %.

## 5. Impact of $O_2$ Columns on $XCO_2$ Measurements

The $O_2$ column retrieved from the 1.27 µm band with a Voigt line shape and spectroscopic parameters from the
atm.101 line list (Toon, G. C., 2014a) has an airmass dependence such that the $O_2$ column retrieved increases as a
function of solar zenith angle (or airmass). Using spectra recorded from Eureka, Park Falls, Lamont, and Darwin
over one-year periods, total columns of $O_2$ were retrieved using (1) a Voigt spectral line shape with spectroscopic
parameters from the atm.101 line list and (2) the qSDV with the spectroscopic parameters determined in Section 3.
Figure 6 shows the percent difference calculated as the column from the qSDV retrieval minus the column from the
Voigt retrieval, which was then divided by the latter and multiplied by 100, plotted as a function of solar zenith
angle (SZA).  At the smallest SZA, the qSDV retrieves 0.75% less $O_2$ than the Voigt, with the difference increasing
to approximately 1.8% as the SZA approaches 90°. Figure 7 shows XAIR for the entire data set plotted as a funtion
of SZA. XAIR is the column of air (determined using surface pressure recorded at the site) divided by the column of
$O_2$ retrieved from the spectra and multiplied by 0.2095, which is the dry air mole fraction of $O_2$ in Earth's
atmosphere. Ideally XAIR should be 1 but when using $O_2$ retrieved with a Voigt line shape (Figure 7a) to calculate
XAIR the average XAIR value for the entire data set is 0.977. Using $O_2$ retrieved with the qSDV, to calculate XAIR
the average value is 0.986 which is closer to the expected value of 1. However, XAIR has a dependence on SZA
regardless of line shape used. Figure 7a shows that XAIR decreases as a function of SZA (evident at SZA> 75°)
which means that the retrieved column of $O_2$ increases as a function of SZA. Figure 7b shows that XAIR increases
as a function of SZA (evident at SZA> 70°), which means that the retrieved column of $O_2$ decreases as a function of
SZA. Therefore retriving total columns of $O_2$ with the qSDV changes the airmass dependendnce of the $O_2$ column
which in turn will impact the airmass dependence of $XCO_2$.

### 5.1 Airmass Dependence of $XCO_2$

Since the standard TCCON $XCO_2$ (and all other XGas) is calculated using the column of $O_2$ instead of the surface
pressure, errors associated with the retrieval of $O_2$, such as the airmass dependence of the $O_2$ column, will affect
$XCO_2$. Figure 8 is $XCO_2$ calculated for four different combinations pertaining to the two $CO_2$ column retrievals and
the $O_2$ column retrievals. The $CO_2$ columns were retrieved with either a Voigt line shape (the standard GGG2014
approach) or the qSDV with line mixing as done in Mendonca et al. (2016) while the $O_2$ columns were retrieved
with either a Voigt (the standard GGG2014 approach) or the new qSDV approach developed here. Figure 8 shows a
spurious symmetric component to $XCO_2$ when the total column of $O_2$ is retrieved with the Voigt line shape,
regardless of line shape used to retrieve $CO_2$. When the qSDV is used to retrieve total columns of $O_2$, the symmetric
component of $XCO_2$ is dismissed regardless of line shape used to retrieve $CO_2$. This is because the airmass
dependence of the column of $O_2$ retrieved using the qSDV is more consistent with the airmass dependence of the
column of $CO_2$ (for both line shapes used to retrieve $CO_2$). Mendonca et al. (2016) showed that using the qSDV with
line mixing results in better fits to the $CO_2$ windows and impacts the airmass dependence of the retrieved column of
$CO_2$. When using a Voigt line shape the retrieved column amount of $CO_2$ decreases as airmass increases until the
airmass is large (SZA of about 82°) at which point the retrieved column of $CO_2$ increases as the airmass increases,
changing the shape of the airmass dependence of the $CO_2$ column. When the qSDV with line mixing is used, the
retrieved column of $CO_2$ decreases as a function of airmass (up until the sun is above the horizon).
To correct for this, an empirical correction is applied to all TCCON $XCO_2$ (and XGas). The empirical correction
determines the antisymmetrical component of the day's $XCO_2$, which is assumed to be the true variation of $XCO_2$
throughout the day, as well as the symmetrical component, which is caused by the airmass dependence of the
retrieved column of the gas of interest and $O_2$. We can, therefore, represent a measurement as (Wunch et al., 2011):

$$y_i = \hat{y}[1 + \alpha S(\theta_i) + \beta A(t_i)] \tag{5}$$

where $\hat{y}$ is the mean value of $XCO_2$ measured that day, $\beta$ is the fitted coefficient of the antisymmetric function $A(t_i)$
and $\alpha$ is the fitted coefficient of the symmetric function $S(\theta_i)$. The antisymmetric function is calculated by (Wunch
et al., 2011):

$$A(t_i) = \sin(2\pi(t_i - t_{noon})) \tag{6}$$

where $t_i$ is the time of the measurement and $t_{noon}$ is the time at solar noon, both in units of days. The symmetric
function is calculated by (Wunch et al., 2011):

$$S(\theta_i) = \left(\frac{\theta_i + 13^o}{90^o + 13^o}\right)^3 - \left(\frac{45^o + 13^o}{90^o + 13^o}\right)^3 \tag{7}$$

where $\theta_i$ is the SZA in degrees. To determine $\alpha$ for the different line shapes, total columns of $CO_2$ were retrieved
using the Voigt line shape (Wunch, D. et al., 2015) and the qSDV with line mixing (Mendonca et al., 2016).
Henceforth, we will refer to $XCO_2$ calculated from $O_2$ and $CO_2$ using the Voigt line shape as $XCO_2$ Voigt and the
qSDV line shape as $XCO_2$ qSDV.
Figure 9 shows the average $\alpha$ calculated for each season at Darwin, Lamont, and Park Falls. Eureka $XCO_2$ cannot be
used to determine $\alpha$ because Eureka measurements do not go through the same range of SZAs as the other three
sites due to its geolocation. The average $\alpha$ values derived from $XCO_2$ Voigt are represented by stars in Figure 9,
while the squares indicate $XCO_2$ qSDV. At all three sites, $\alpha$ is closer to 0 when the qSDV line shape is used in the
retrieval compared to the Voigt retrieval, regardless of the season. The average $\alpha$ for $XCO_2$ Voigt calculated from a
year of measurements from Darwin, Park Falls, and Lamont is -0.0071±0.0057 and that for $XCO_2$ qSDV is -

0.0012±0.0054.

For all four sites, $\alpha = $ -0.0071 is used to correct $XCO_2$ Voigt measurements. Figure 10a shows the $XCO_2$ Voigt
anomalies plotted as a function of SZA. The data is expressed as the daily $XCO_2$ anomaly, which is the difference
between the $XCO_2$ value and the daily median value, in order to remove the seasonal cycle. When $XCO_2$ is left
uncorrected for airmass dependencies, $XCO_2$ decreases as a function of SZA up to approximately 82°, and increases
as a function of SZA at angles greater than 82°. Figure 10b shows $XCO_2$ Voigt corrected for the airmass
dependence. This airmass correction works well only up to a SZA of approximately 82°. Figure 10c is the same as
10a but for the uncorrected $XCO_2$ qSDV measurements, while Figure 10d is the same as 10b but for the corrected
$XCO_2$ qSDV measurements. When the airmass correction is applied to $XCO_2$ qSDV there is a small difference
between the corrected and uncorrected $XCO_2$ qSDV measurements, with the difference only noticeable for the
Darwin measurements recorded at SZA > 60°. For $XCO_2$ qSDV measurements made at SZA > 82° $XCO_2$ does not
increase with SZA as it does with the Voigt.
**5.2 Accuracy of $XCO_2$**
To assess the accuracy of TCCON $XCO_2$ measurements, they are compared to aircraft $XCO_2$ profile measurements
using the method described in Wunch et al. (2010). Figure 11a shows the comparison between the aircraft $XCO_2$
(Deutscher et al., 2010; Lin et al., 2006; Messerschmidt et al., 2010; Singh et al., 2006; Wofsy, 2011) measurements
(legend at the top details the different aircraft) and TCCON $XCO_2$ Voigt measurements for 13 TCCON sites (given
by the color-coded legend at the bottom right). The gray line indicates the one-to-one line and the dashed line is the
line of best fit. There is a bias of 0.9897±0.0005, as given by the slope of the line of best fit in Figure 11a, for the
$XCO_2$ Voigt measurements. Figure 11b is the same as 11a but for the $XCO_2$ qSDV measurements. The bias between
the aircraft $XCO_2$ measurements and the $XCO_2$ qSDV measurements is 1.0041±0.0005 as given by the slope of the
line of best fit in Figure 11b. This increase in the slope can be explained by an increase in the retrieved column of
$CO_2$ when using the qSDV with line mixing as shown in Mendonca et al. (2016) as well as combined with a
decrease in the retrieved $O_2$ column due to using the qSDV. As discussed previously (section 5) the decrease in the
retrieved $O_2$ column is an improvement but the expected column of $O_2$ is still approximately 1.2% higher (at the
smallest SZA) than it should be. Therefore, the retrieved column of $CO_2$ is higher than it should be, and the slope
would be greater if the retrieved column of $O_2$ was 1.2% lower. Never the less using the qSDV to retrieve total
columns of $CO_2$ and $O_2$ reduces the difference between TCCON $XCO_2$ and aircraft $XCO_2$ measurements by 0.62 %.
TCCON $XCO_2$ measurements are divided by the scale factors (or bias determined in Figure 11) to calibrate to the
WMO scale. For all TCCON $XCO_2$ measurements retrieved with a Voigt line shape, the airmass correction is first
applied to the data and the result is divided by the determined bias factor, 0.9897. Figure 12a to 12d shows $XCO_2$
Voigt (for Eureka, Park Falls, Lamont, and Darwin respectively) indicated by red square boxes in the plots. $XCO_2$
Voigt measurements made at SZA > 82° have been filtered out because they cannot be corrected for the airmass
dependence. The blue boxes are $XCO_2$ qSDV corrected for airmass dependence and scaled by 1.0041. No filter was
applied to the $XCO_2$ qSDV measurements for SZA since the airmass dependence correction works at all SZA.
Figure 12e to 12h shows the difference between $XCO_2$ Voigt and $XCO_2$ qSDV for Eureka, Park Falls, Lamont, and
Darwin respectively. The mean differences for the data shown in Figures 12e to 12h are 0.113±0.082, -0.102±0.223,
-0.132±0.241, and -0.059±0.231 µmol/mol (ppm) for Eureka, Park Falls, Lamont, and Darwin respectively. The
difference throughout the day at Park Falls, Lamont, and Darwin varies between -0.6 to 0.2 μmol/mol and is SZA
dependent.
Figure 13a shows $XCO_2$ Voigt corrected for the airmass dependence, as well as $XCO_2$ qSDV, uncorrected and
corrected for the airmass dependence. These $XCO_2$ measurements were retrieved from Park Falls spectra recorded
on June 18, 2013. For all three $XCO_2$ measurements, the amount of $XCO_2$ decreases throughout the day. Figure 13b
shows the difference between the corrected Voigt $XCO_2$ and the uncorrected qSDV $XCO_2$, as well as the difference
between the corrected Voigt $XCO_2$ and the corrected qSDV $XCO_2$. The difference between the Voigt and the qSDV
(corrected and uncorrected) shows that at the start and end of the day, more $XCO_2$ is retrieved with the qSDV, while
at midday less is retrieved with the qSDV. The range in the differences seen in Figure 12e to 12h varies with SZA
throughout the day as shown in Figure 13b.
**6. Discussion and Conclusions**
Using cavity ring-down spectra measured in the lab, we have shown that the Voigt line shape is insufficient to
model the line shape of $O_2$ for the 1.27 μm band, consistent with the results of (Hartmann et al. (2013) and
Lamouroux et al. (2014). By using the speed-dependent Voigt line shape when calculating the absorption
coefficients, we were better able to reproduce the measured absorption coefficients than using the Voigt line shape.
However, some residual structure remains as seen Figures 1 and 2. This is partly due to the blending of spectral lines
(i.e., line mixing) and the inability to retrieve the spectroscopic parameters for weak $O_2$ transitions. Fitting low-
pressure spectra would help with isolating spectral lines and decreasing the uncertainty on the retrieved
spectroscopic parameters for the Q branch lines.
Accurate measurements of the pressure shifts in the 1.27 μm band have been hard to obtain as shown in Newman et
al. (1999) and Hill et al., (2003). While the retrieved pressure shifts show a dependence on quantum number m
(Figure 3c) as one would expect, this dependence is not as strong as the m dependence of the Lorentz widths (Figure
3b). This can be explained by the fact that line mixing, which is shown to be important for the $O_2$ A-band, was not
considered when fitting the cavity-ringdown spectra. Neglecting line mixing usually produces an asymmetric
residual in the discrete lines as well as a broad residual feature associated with the fact that collisions are transferring
intensity from one part of the spectrum to another. By fitting a set of Legendre polynomials for CIA we could be
simultaneously fitting the broader band feature associated with line mixing while the retrieved pressure shifts, and
speed-dependent pressure shifts could be compensating for the asymmetric structure one would see in the discrete
lines when neglecting line mixing. The remaining structure, as seen in Figure 1c, could be due to neglecting line
mixing especially in the Q-branch where the spacing between spectral lines is small (in comparison to the P and R
branches) and line mixing is most likely prevalent. The large error bars for the measured pressure shifts and speed-
dependent pressure shifts as well as a deviation from a smooth m dependence of these parameters could be due to
neglecting line mixing when fitting the lab spectra. Figure 3c and 3d show that the spectral lines that have large error
bars and deviate from an expected m dependence belong mainly to the Q-branch spectral lines (which are mostly
likely impacted by line mixing). To achieve the results obtained in this study it is best to use the parameters as is
instead of trying to apply an interpolation, that depends on m, or even omitting them unless one test's these changes
on atmospheric spectra that cover different range of conditions (i.e. seasons, dry/wet, SZA, geographical locations).
It is evident that the parameters might be compensating for affects (such as line mixing) that were not included when
fitting the lab spectra and changing these parameters (or omitting them) could lead to degradation in the quality of
the spectral fits of solar spectra and change the airmass dependence of the retrieved column of $O_2$ which would
impact the airmass dependence of $XCO_2$.
The pressure dependence of the retrieved speed-dependent width parameter is an indication that Dicke narrowing
needs to be taken into account, as shown by Bui et al. (2014) for $CO_2$. When both speed dependence and Dicke
narrowing are present, a multi-spectrum fit needs to be used due to the correlation between the parameters (Bui et
al., 2014). Domysławska et al. (2016) recommend using the qSDV to model the line shape of $O_2$ based on multiple
line shape studies of the $O_2$ B-band. In these studies, a multi-spectrum fit to low pressure (0.27-5.87 kPa) cavity-ring
down spectra was preformed testing multiple line shapes that took speed-dependence and Dicke narrowing into
account both separately and simultaneously. They found that the line shapes that only used Dicke narrowing were
not good enough to model the line shape of the $O_2$ B-band lines, but a line shape that included either speed-
dependence or both speed-dependence and Dicke narrowing produced similar quality fits, ultimately concluding that
speed-dependence has a larger effect than Dicke narrowing. It was noted in the study by Wójtewicz et al., (2014)
that both Dicke narrowing and speed-dependent effects might simultaneously play an important role in modeling the
line shape of the $O_2$ B-band lines. However, the speed-dependent and Dicke narrowing parameters are highly
correlated at low pressures. To reduce the correlation requires either a multi-spectrum fit of spectra at low pressures
with high enough signal to nosie ratio or spectra that cover a wide range of pressure (Wójtewicz et al., 2014). So, by
combining the high-pressure spectra used in this study with low pressure spectra in a multispectrum fit both the
speed-dependence and Dicke narrowing parameters could be retrieved. The temperature dependence of the Lorentz
width coefficients of this band has never been measured before, which could have an impact on the airmass
dependence of $O_2$. Combining high-pressure cavity-ring-down absorption coefficient measurements with those for
low pressures and different temperatures as done in Devi et al. (2015 and 2016) for $CH_4$ would lead to more accurate
line shape parameters for $O_2$.
By taking speed dependence into account for both $CO_2$ (in the work of Mendonca et al., 2016) and $O_2$ (the work
presented here), we were able to significantly decrease the airmass dependence of TCCON $XCO_2$ and the bias
between TCCON and aircraft $XCO_2$. XAIR calculated with the column of $O_2$ retrieved with the qSDV is now closer
to the expected value of 1 but XAIR still has an airmass dependence which is the results of the retrieved total
column of $O_2$ decreasing as a function of SZA at large SZA. This remaining airmass dependence could be due to
neglecting affects such as Dicke narrowing and line mixing in the absorption coefficient calculations, as well as
assuming a perfect instrument line shape in the retrieval algorithm. However, retrieving $O_2$ with the qSDV
significantly decreases the airmass dependence of $XCO_2$. With the qSDV line shape, $XCO_2$ measurements made at
SZA > 82° no longer have to be discarded. We recommend using the full range of SZA which would result in more
$XCO_2$ measurement available from all TCCON sites. This is particularly important for high-latitude TCCON sites,

such as Eureka, because measurements made from late February to late March and from late September until mid-October are made at SZA > 82°. Filtering out these large SZA measurements thus limits the knowledge of the seasonal cycle of $XCO_2$ at high latitudes. The airmass dependence of the $O_2$ column not only effects $XCO_2$ but all trace gases measured by TCCON and in the future the airmass dependence of all XGas will be determined with these new $O_2$ columns.

**Acknowledgements**

This work was primarily supported by the Canadian Space Agency (CSA) through the GOSAT and CAFTON projects and the Natural Sciences and Engineering Research Council of Canada (NSERC). The Eureka measurements were made at the Polar Environment Atmospheric Research Laboratory (PEARL) by the Canadian Network for the Detection of Atmospheric Change (CANDAC), which has been supported by the AIF/NSRIT, CFI, CFCAS, CSA, Environment Canada (EC), Government of Canada IPY funding, NSERC, OIT, ORF, PCSP, and FQRNT. The authors wish to thank the staff at EC's Eureka Weather Station and CANDAC for the logistical and on-site support provided. Thanks to CANDAC Principal Investigator James R. Drummond, PEARL Site Manager Pierre Fogal, and CANDAC/PEARL operators Mike Maurice and Peter McGovern, for their invaluable assistance in maintaining and operating the Bruker 125HR. The research at the Jet Propulsion Laboratory (JPL), and California Institute of Technology was performed under contracts and cooperative agreements with the National Aeronautics and Space Administration (NASA). Geoff Toon and Debra Wunch acknowledge support from NASA for the development of TCCON via grant number NNX17AE15G. Darwin TCCON measurements are possible thanks to support from NASA grants NAG5-12247 and NNG05-GD07G, the Australian Research Council grants DP140101552, DP110103118, DP0879468 and LP0562346, and the DOE ARM program for technical support. The research at the National Institute of Standards and Technology was performed with the support of the NIST Greenhouse Gas Measurements and Climate Research Program. Certain commercial equipment, instruments, or materials are identified in this paper in order to specify the experimental procedure adequately. Such identification is not intended to imply recommendation or endorsement by the National Institute of Standards and Technology, nor is it intended to imply that the materials or equipment identified are necessarily the best available for the purpose.

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

**Figures**

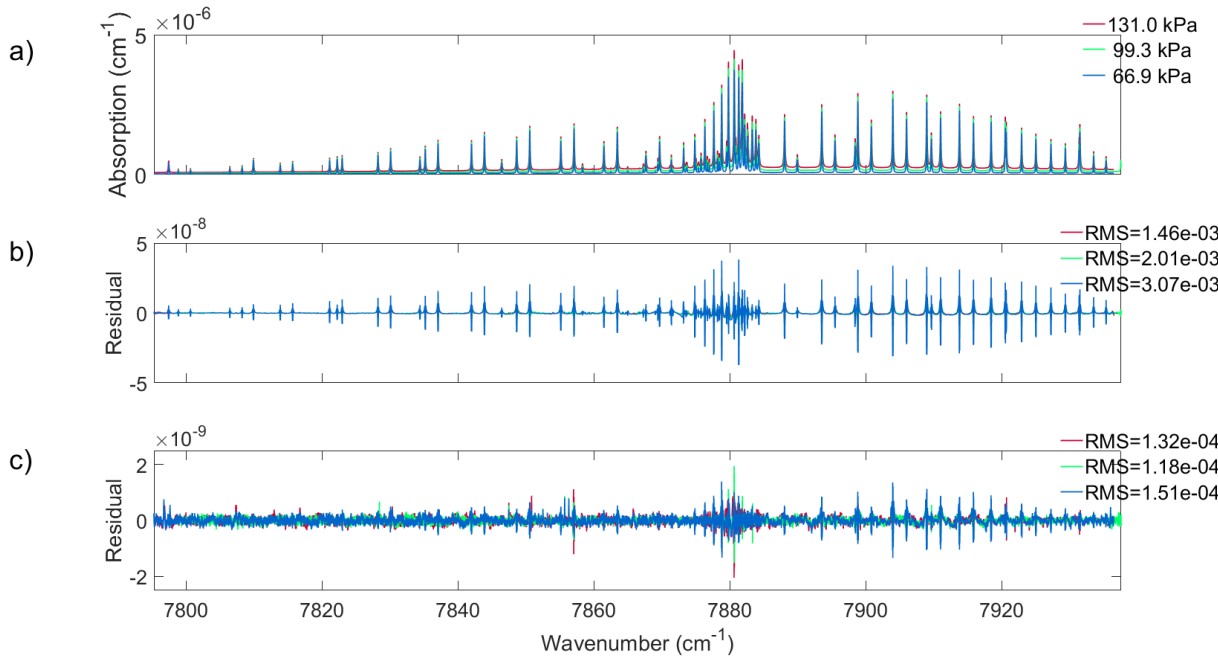


**Figure 1: (a) Cavity-ring-down absorption coefficients for O$_2$ measured at the three pressures indicated in the**
**legend at approximately room temperature and a volume mixing ratio of 0.20720(43). The difference between**
**measured absorption coeffcients and those calculated using (b) a Voigt line shape, and (c) the speed-**
**dependent Voigt line shape. Note the difference in scale between panels (b) and (c).**



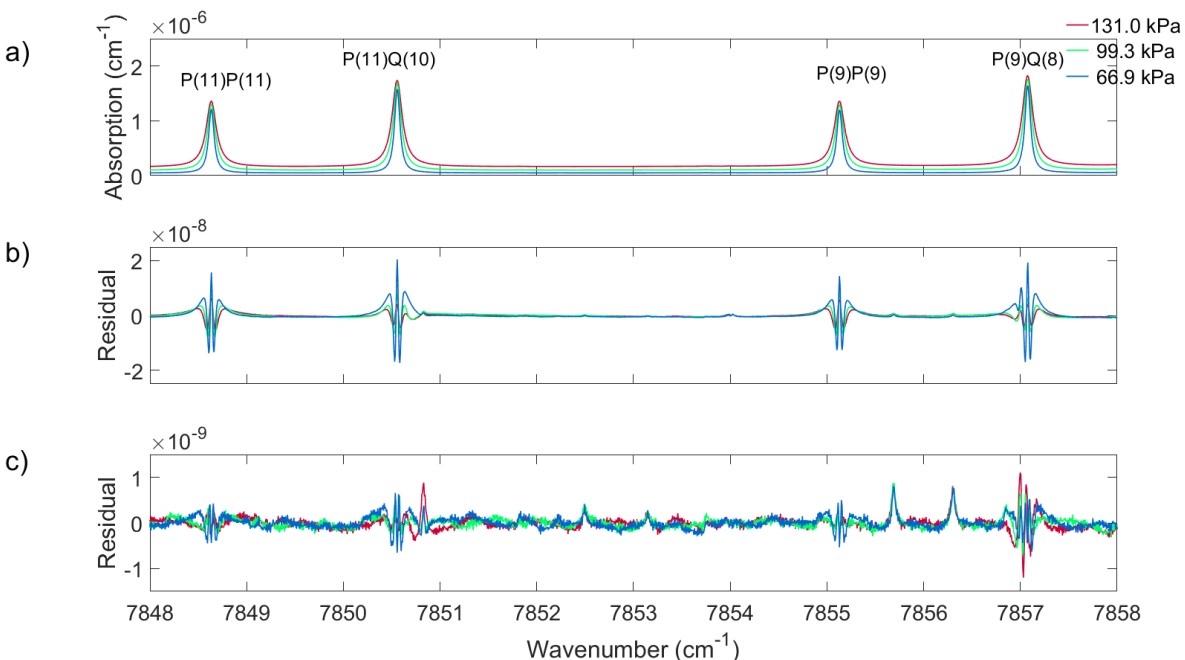


**Figure 2: The same as Figure 1 but expanded to show four spectral lines in the P branch of the $O_2$ 1.27 µm**
**band.**







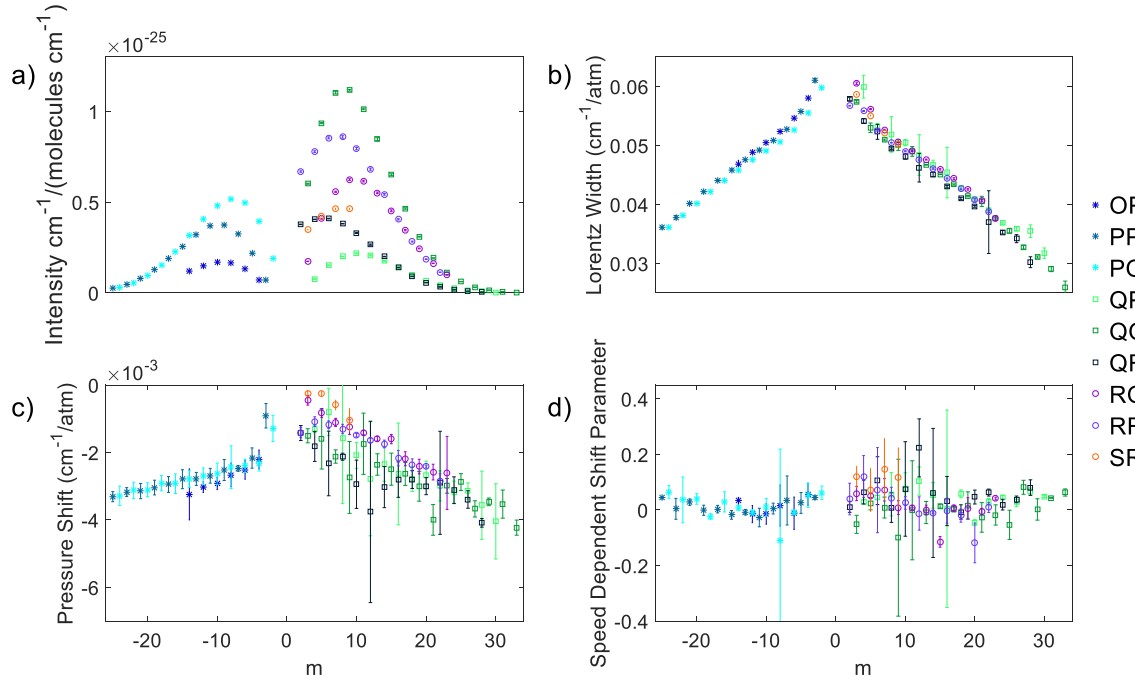


**Figure 3: The averaged measured (a) intensity, (b) Lorentz line width, (c) pressure shift, and (d) speed-**
**dependent pressure shift retrieved from the three cavity ring-down spectra of the 1.27 μm band of O₂. All**
**data are plotted as a function of m which is m =-J for the P-branch lines, m=J for the Q-branch, and m=J+1**
**for the R-branch (where J is the lower state rotational quantum number) and the uncertainties shown are 2σ.**







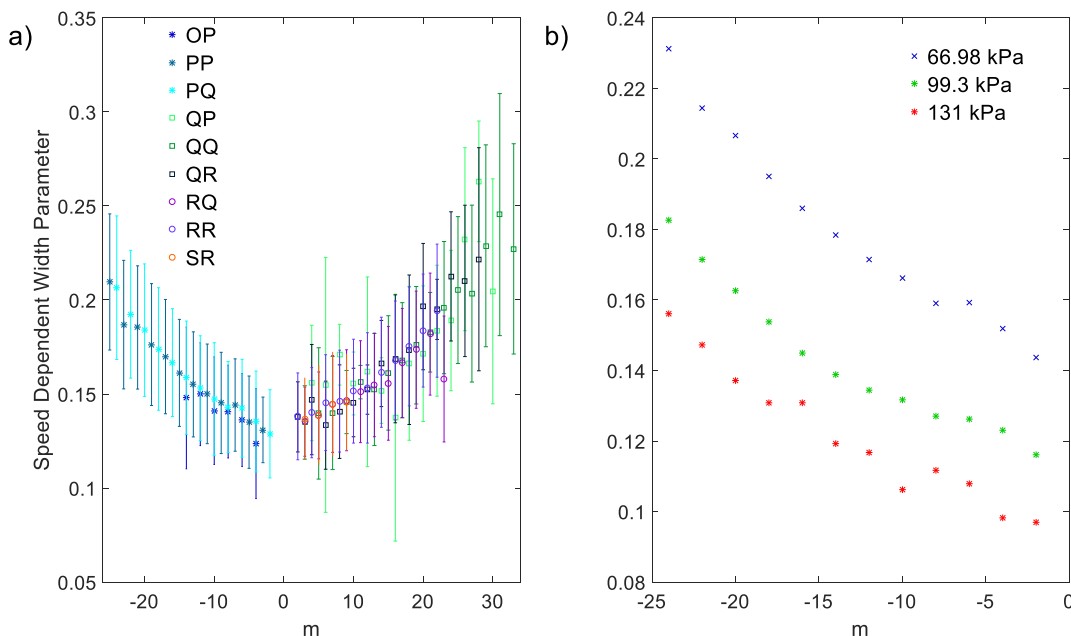


Figure 4: (a) The averaged measured speed-dependent width parameter of the 1.27 μm band of $O_2$ plotted as a function of m. (b) The measured speed-dependent width parameter for spectral lines that belong to the PQ sub-branch plotted as a function of m.





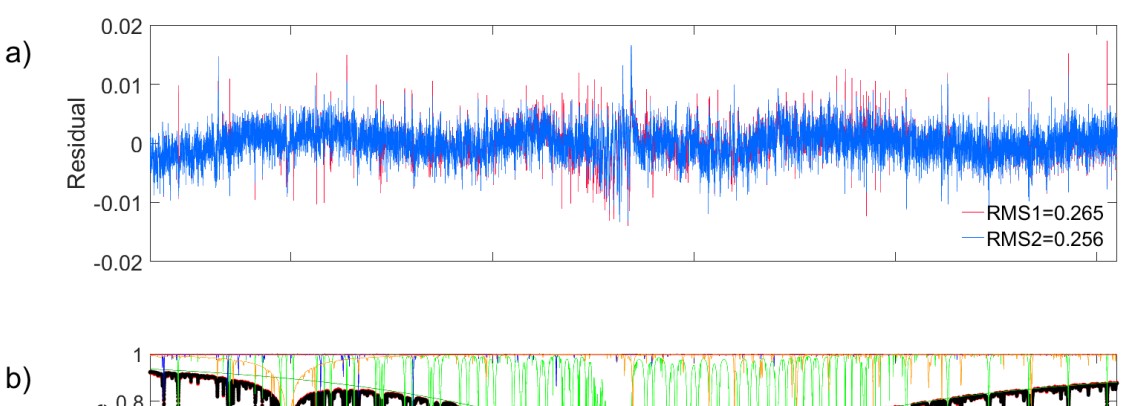

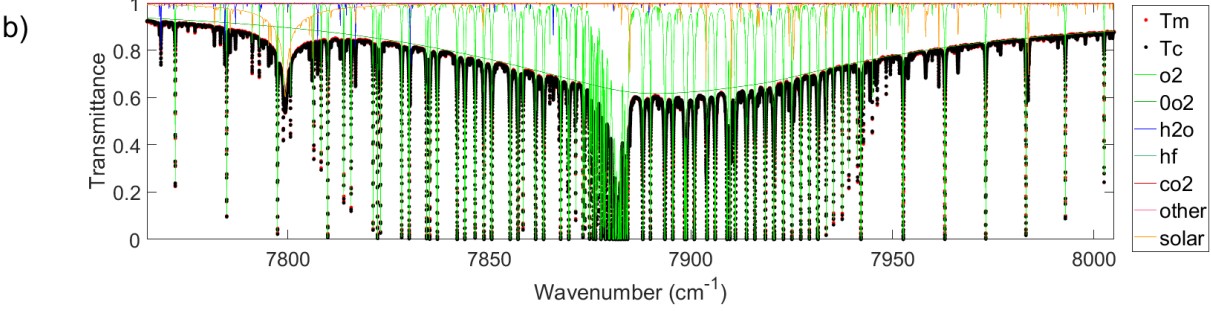


**Figure 5: (a) The residuals (measured minus calculated) for a spectrum measured at Eureka on March 27,**
**2015 at a SZA of 81.32º. The red residual is the result of using the Voigt line shape and the blue is from using**
**the qSDV. (b) The measured (red dots) and calculated (black dots), with the qSDV, spectrum, along with the**
**gases included in the fit (refer to the legend to the right) in the spectral window.**






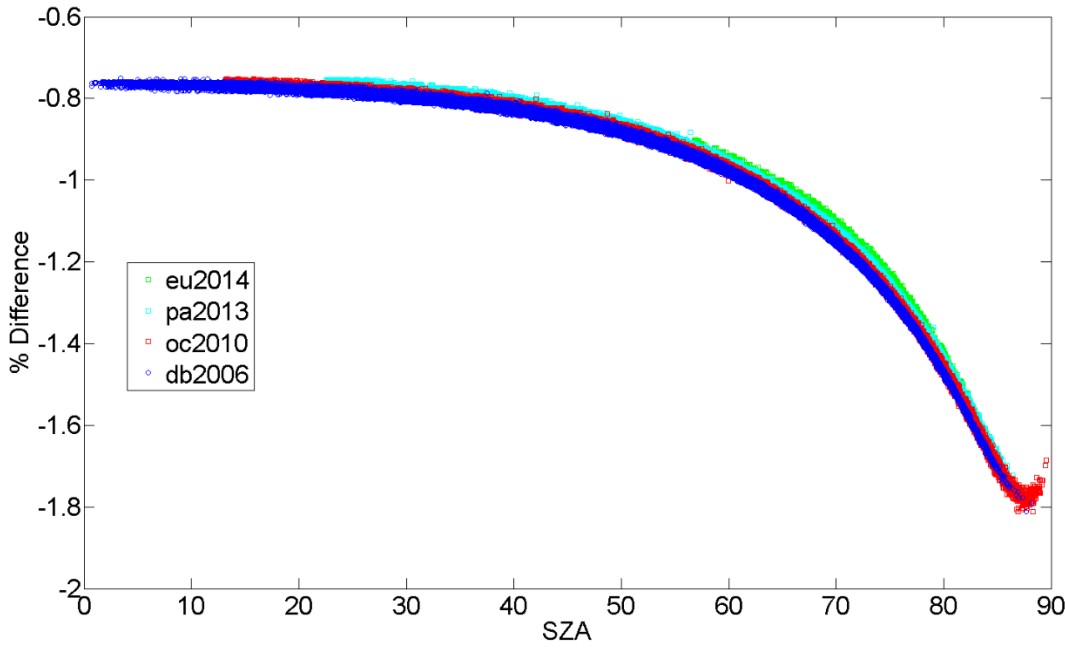


**Figure 6: The percent difference between the O$_2$ column retrieved with the Voigt and qSDV line shapes for a year of measurements from Eureka (eu), Park Falls (pa), Lamont (oc), and Darwin (db).**





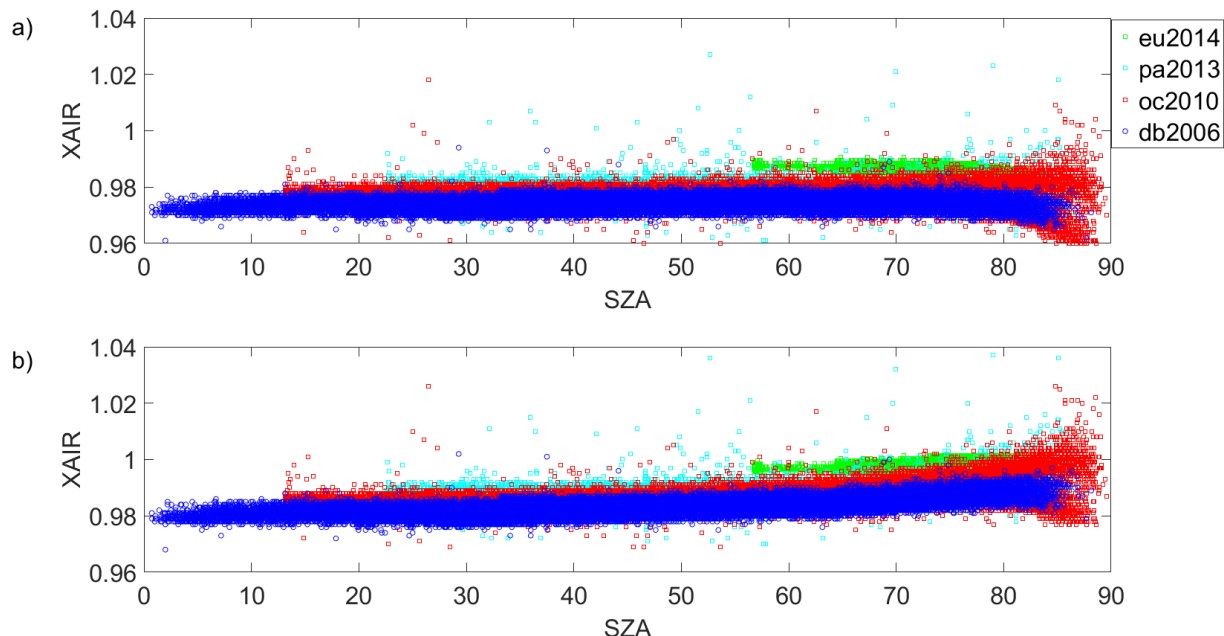


**Figure 7: (a) XAIR as a function of SZA calculated using the total column of $O_2$ retrieved using the Voigt line shape. (b) is the same as (a) except the total column of $O_2$ was retrieved with the qSDV.**





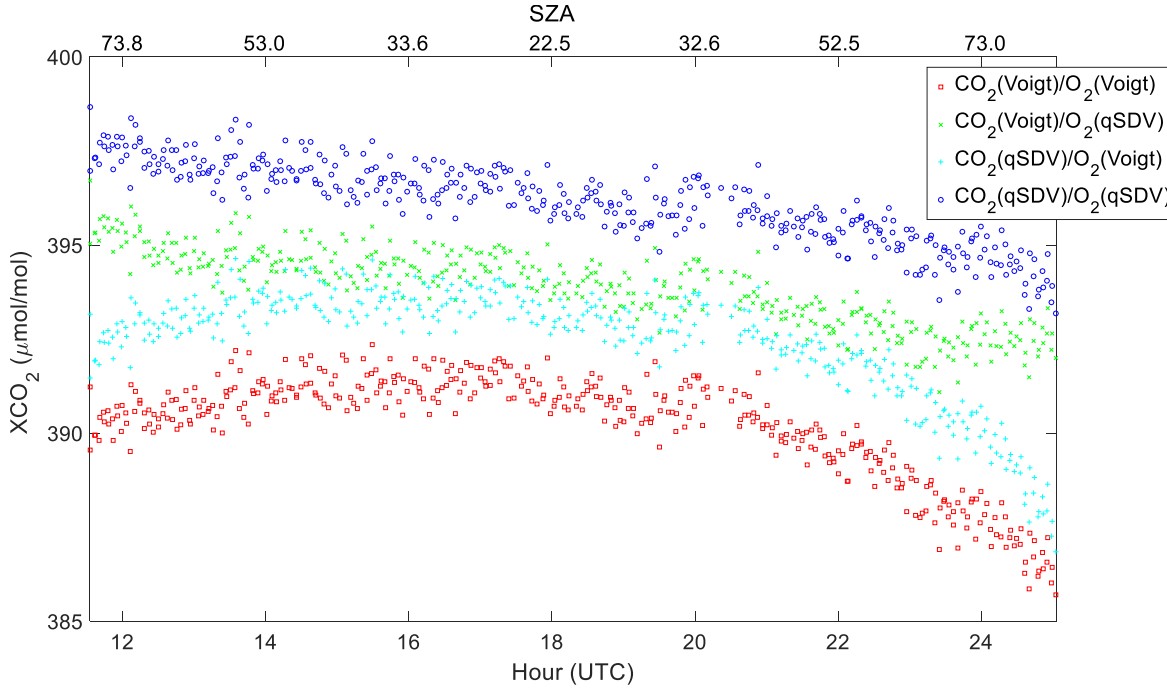


**Figure 8: XCO₂ calculated from the CO₂ and O₂ columns retrieved from Park Falls spectra recorded on June 18, 2013. The CO₂ columns were retrievied using either the Voigt line shape or the qSDV with line mixing, while the O₂ columns were retrieved using either the Voigt or qSDV line shapes. XCO₂ was calculated in four ways: 1) Both CO₂ and O₂ columns retrieved using the Voigt line shape (red), 2) CO₂ columns retrieved with the Voigt and O₂ columns retrieved with the qSDV (green), 3) CO₂ columns retrieved with the qSDV and line mixing and O₂ columns retrieved with the Voigt (cyan), and 4) CO₂ columns retrieved with the qSDV and line mixing and O₂ columns retrieved with the qSDV (blue). The top x-axis is the SZA that corresponds to the hour on the bottom x-axis.**



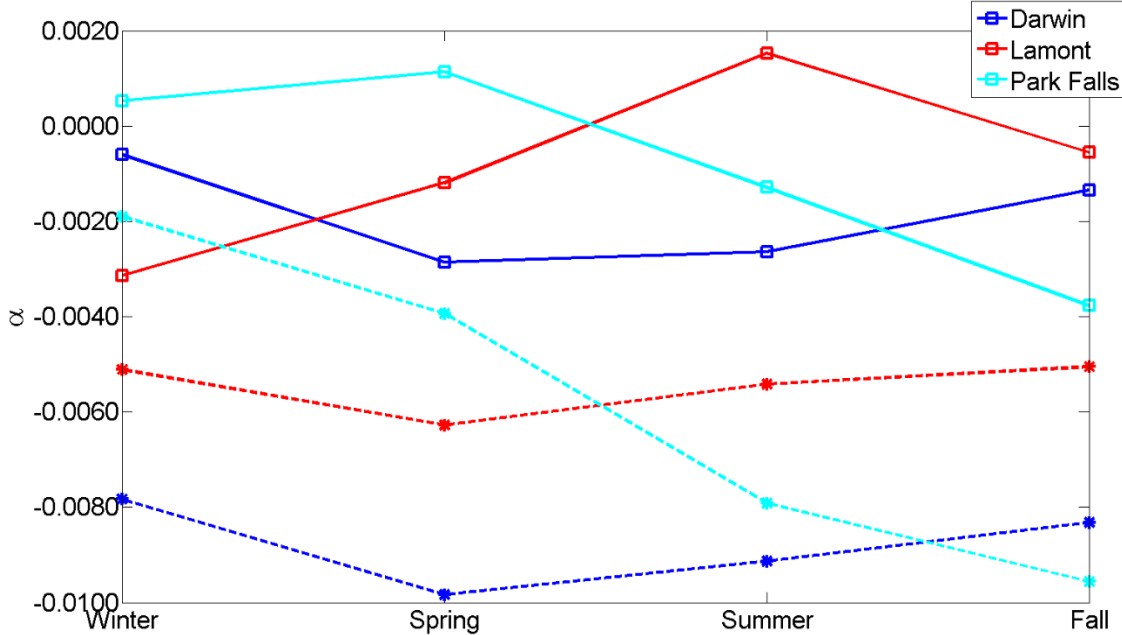


**Figure 9: The average airmass-dependent correction factor for XCO₂ derived from a year of spectra**
**measured at Darwin, Lamont, and Park Falls for different seasons. The dashed lines with stars are the α for**
**XCO₂ retrieved using a Voigt line shape for both CO₂ and O₂ columns. The solid lines with squares are from**
**XCO₂ retrieved using the qSDV for both CO₂ and O₂ columns.**







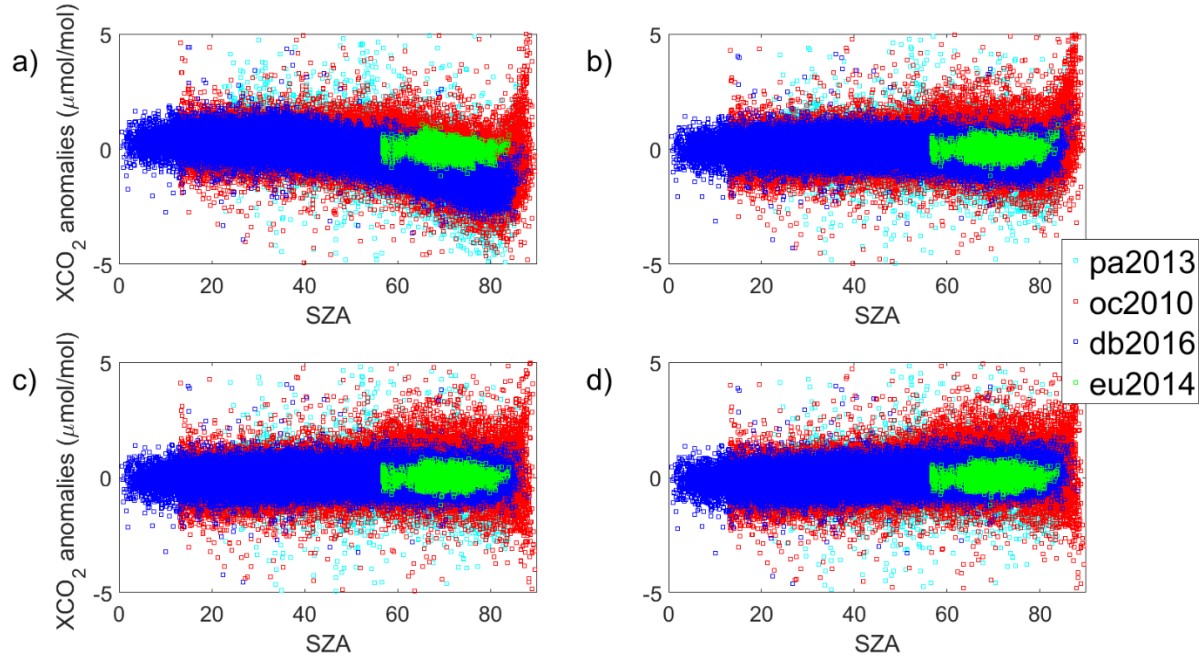


**Figure 10: (a) XCO₂ Voigt anomaly for a year of measurements from the four TCCON sites. The XCO₂**
**anomaly is the difference between each XCO₂ value and the daily median XCO₂. (b) The XCO₂ Voigt**
**anomaly after the airmass dependence correction is applied to the XCO₂ Voigt data. (c) XCO₂ qSDV**
**anomaly. (d) XCO₂ qSDV anomaly after correction for the airmass dependence.**






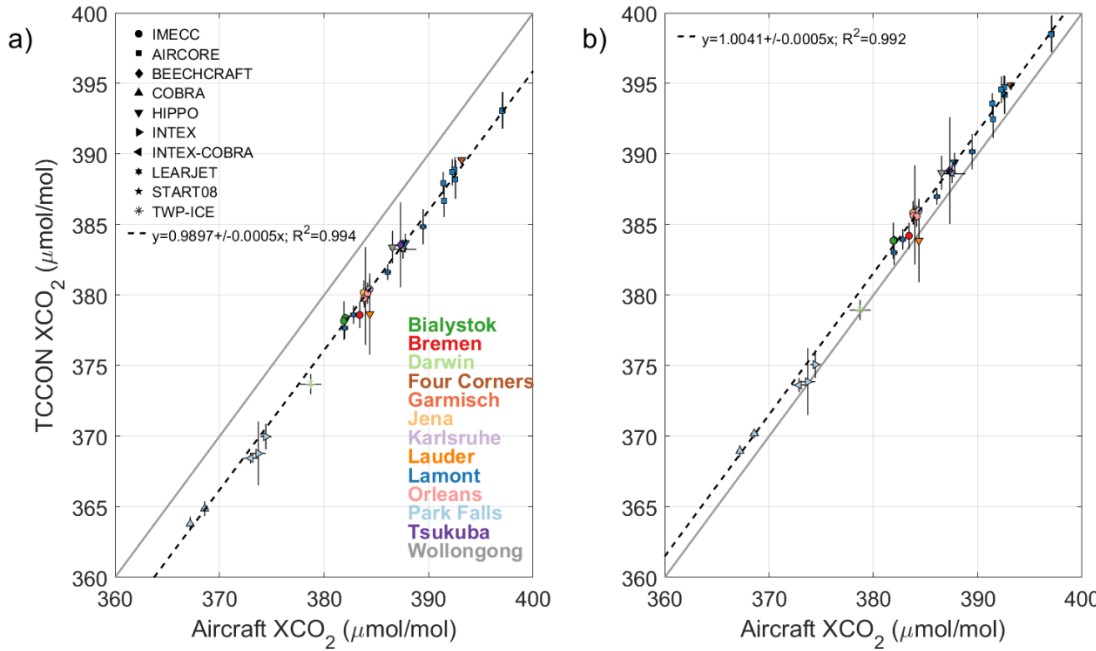


**Figure 11: (a) Correlation between TCCON and aircraft XCO₂ Voigt measurements for 13 TCCON sites.**
**Each aircraft type is indicated by a different symbol given by the legend in the top left corner. Each site is**
**represented by a different colour given by the legend in the bottom right corner. The grey line indicates the**
**one-to-one line and the dashed line is the line of best fit for the data. The slope of the line of best fit as well as**
**the error on the slope are given in the plot. (b) the same as (a) but for XCO₂ qSDV.**




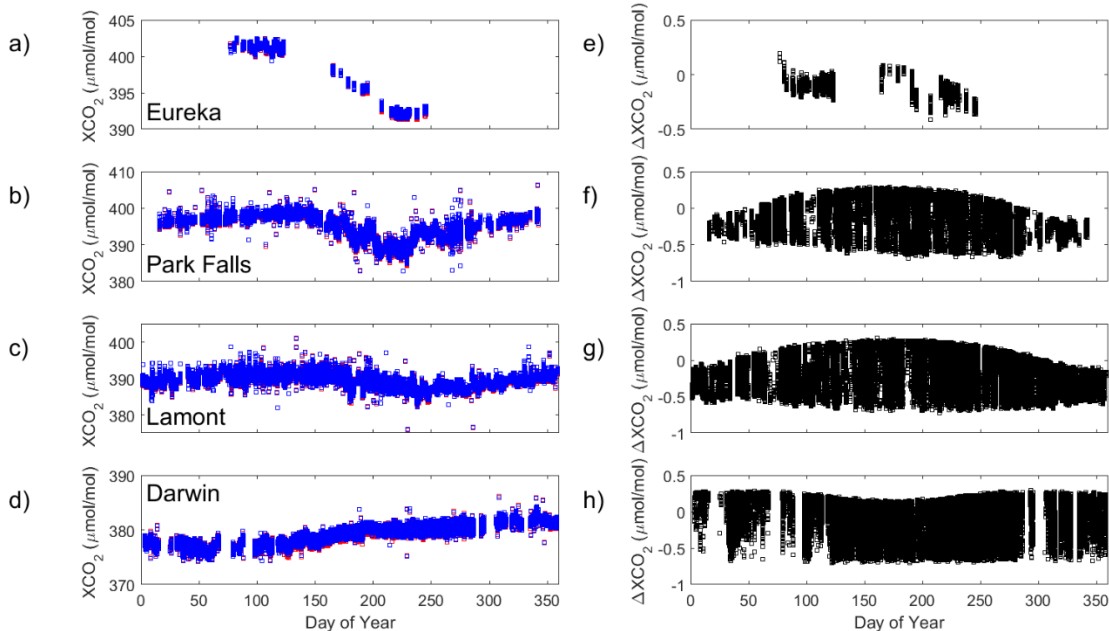


**Figure 12: (a) to (d) XCO₂ plotted as a function of day of the year for Eureka (2014), Park Falls (2013),**
**Lamont (2010), and Darwin (2006) respectively. The mostly-hidden red boxes are XCO₂ calculated from**
**using a Voigt line shape in the retrieval and the blue boxes are from using the qSDV. (e) to (h) the difference**
**between XCO₂ Voigt and XCO₂ qSDV.**




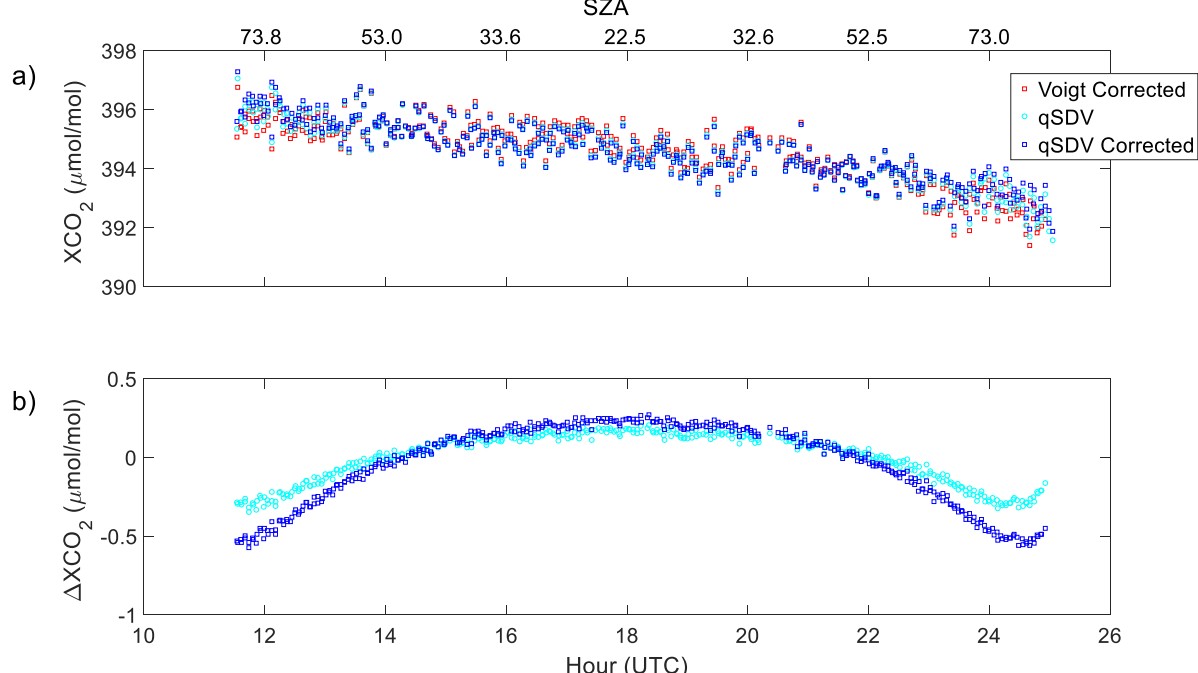


**Figure 13: (a) XCO₂ from Park Falls retrieved from spectra recorded on June 18, 2013. Plotted is XCO₂**
**retrieved: (1) with a Voigt line shape and corrected for the airmass dependence (red squares), (2) with the**
**qSDV (cyan circles), and (3) with the qSDV and corrected for the airmass dependence (blue squares). (b) the**
**difference between the Voigt XCO₂ corrected and the qSDV XCO₂ (cyan circles), and the difference between**
**the Voigt XCO₂ corrected and the qSDV XCO₂ corrected (blue squares). The top x-axis is the SZA that**
**corresponds to the hour on the bottom x-axis.**