# Peer review of "Measurements of XCO2"

_Atmospheric Measurement Techniques, 2018_

## Referee Comment (RC1) · Anonymous Referee #1 · 8 Jun 2018

This is a very good paper describing accurate measurements of spectral parameters of the magnetic dipole lines in the 1.27 micron band of molecular oxygen. Authors used speed-dependent Voigt (SDV) line shape which is known to produce more accurate results when applied at terrestrial atmospheric conditions. The retrieved parameters were then efficiently used in application to the atmospheric TCCON measurements. The improved line list required smaller empirical correction factors with respect to the previous linelists including HITRAN2012. It is commendable that authors made their line list available. I think the paper should definitely be published after following comments are addressed:

1. In lines 55-62 authors talk about different works on the A-band and mentioning Galatry retrievals. However, for some reason the work of Drouin et al (JQSRT 2016)

was not mentioned although it also employed the SDV profile.

2. The authors may want to mention that the HITRAN2016 parameters are very similar to those in HITRAN2012 in this particular band and the only change are improved line positions from Yu et J. Chem. Phys. 141 (2014) 174302. doi:10.1063/1.4900510.

3. It is interesting that the authors do mention the line-mixing with respect to the A-band and CO2 bands but did not say about this effect in 1.27 micron band that they investigated. It is also not mentioned as potential source of remaining residuals in lines 293-303. It would be interesting to see some discussion about this.

4. Talking about the sources of the residuals and its potential relation to Dicke narrowing it would be interesting what authors think about conclusions of the Torun group (Domyslawska et al papers in JQSRT 2014-2016), that for the electronic transitions of O2 speed-dependence should have much larger effect than Dicke narrowing.

5. Spectral shifts in the 1.27 micron band had always been very hard to measure. See for instance discussion in Hill et al, J. Mol. Spectrosc. 221 (2003) 286–287. doi:10.1016/S0022-2852(03)00227-3 and Newman, et al, J. Chem. Phys. 110 (1999) 10749. doi:10.1063/1.479018.

The authors may want to mention this. Continuing the topic of shifts it is well known that while the widths in P and R branches for same rotational quanta should be very similar the shifts should be assymetric. Therefore I would suggest to plot these separately or using running number m, where m=-J for P lines and J+1 for R. The authors may also want to use the upper state rotational quanta because they are not split into spin components.

6. Does one need to account for airglow when analysing the 1.27 TCCON spectra? See Sun et al (https://doi.org/10.1029/2018GL077823) for instance, regarding significance of airglow in oxygen's 1.27 micron band at the top of atmosphere.

---

## Referee Comment (RC2) · Anonymous Referee #2 · 11 Jun 2018

This is a very nice and important paper. It should be published in AMT, I have only a few minor comments.

1. The paper is very theoretical. Line 92 starts with: to take speed dependence into account ... Here it would be nice to explain what is meant by speed dependence. It should be mentioned that the assumed basis for the Lorentz portion of the Voigt profile is, that for all collisions between the molecules the statistical average velocity is taken. However, in reality this is not true, the molecules have a distribution of speeds, which requires the qSDV.

2. The same holds for the Dicke narrowing, mentioned in line 59. What is the Dicke narrowing? It should be mentioned that when the mean free path of an atom is much smaller than the wavelength of the radiative transition, the atom changes velocity and

direction many times during the emission or absorption of a photon. This causes an averaging over different Doppler states and results in an atomic linewidth that is narrower than the Doppler width (I have taken this from Wikipedia).

3. The $O_2$ concentration in the atmosphere is very stable and well known. I would be interested to see the difference between the known $O_2$ concentration and the $O_2$ from the TCCON spectra as a function of the SZA. These results are somehow hidden in the paper (Figure 6), but since the qSDV is applied to $CO_2$ and $O_2$ it would be good to see where the differences mentioned (0.004) are coming from, from $CO_2$ or $O_2$.

4. I found it a bit disappointing that the airmass dependance is now + 0.004 instead of - 0.013. This is a large reduction, but the results show that still something is wrong in the measurements/retrieval. The authors might discuss this in more detail. See above at 3.

5. For me the fact that the airmass dependence is nearly gone when applying qSDV (Figure 8) very important. This should be more highlighted as main result. Figure 8 c and d look very similar. For me an airmass correction is not necessary, or is this a mistake in the panels?

6. May be a Figure showing $XCO_2/O_2$ as a function of SZA for i) $XCO_2/O_2$, ii) $XCO_2(sQDV)/O_2$ ii) $XCO_2/O_2(sQDV)$, iii) $XCO_2 (qSDV)/O_2(qSDV)$ would be interesting to see where the improvement is coming from. For me a few other Figures of 1-5 could be deleted or put in the supplement.

7. The main part of the paper deals with the speed-dependent Voigt line shape. I would suggest to include this in the title, may by: Improving the Retrieval of $XCO_2$ from Total Carbon Column Network Solar Spectra by inclusion of the speed-dependent Voigt line shape.

6. In the conclusions the authors write: Using cavity ring-down spectra measured in the lab, we have shown that the Voigt line shape is insufficient to 290 model the line

shape of O2 for the 1.27 $\mu$m band, ... As far as I see, the improvement might also results because the qSDV is applied also the $CO_2$.

---

## Author Comment (AC2) · 20 Jul 2018

Response to Referee 2

We thank the reviewer for the comments on our manuscript. Attached is the updated manuscript with all changes. In the manuscript highlighted text is the added text and red crossed out text is deleted text. Please see below for our response to the comments.

Comment 1 - The paper is very theoretical. Line 92 starts with: to take speed dependence into account ... Here it would be nice to explain what is meant by speed dependence. It should be mentioned that the assumed basis for the Lorentz portion of the Voigt profile is, that for all collisions between the molecules the statistical average velocity is taken. However, in reality this is not true, the molecules have a distribution of speeds, which requires the qSDV.

To address this, we have added the following:

Lines 104-105: "The Voigt line shape is the convolution of the Lorentz and the Gaussian profiles, which model pressure and doppler broadening of the spectral line respectively."

Lines 112-113: "The Voigt line shape assumes that pressure broadening is accurately represented by a Lorentz profile calculated for the stastical average velocity at the time of collission."

Lines 115-118: "The speed-dependent Voigt line shape refines the pressure broadening component of the Voigt by calculating multiple Lorentz profiles for different speeds at the time of collision. The final contribution from pressure broadening to the speed-dependent Voigt is the weighted sum of Lorentz profiles (weighted by the Maxwell-Boltzmann speed-distribution) calculated for different speeds at the time of collision."

Comment 2 - The same holds for the Dicke narrowing, mentioned in line 59. What is the Dicke narrowing? It should be mentioned that when the mean free path of an atom is much smaller than the wavelength of the radiative transition, the atom changes velocity and direction many times during the emission or absorption of a photon. This causes an averaging over different Doppler states and results in an atomic linewidth that is narrower than the Doppler width (I have taken this from Wikipedia).

To address this comment we have added the following:

Lines 71-74: "Dicke narrowing occurs when the motion of the molecule is diffusive due to collisions changing the velocity and direction of the molecule during the time that it is excited. This diffusive motion is taken into account by averaging over many different Doppler states resulting in a line width that is narrower than the Doppler width (Dicke, 1953)."

Comment 3 - The O2 concentration in the atmosphere is very stable and well known.

I would be interested to see the difference between the known O2 concentration and the O2 from the TCCON spectra as a function of the SZA. These results are somehow hidden in the paper (Figure 6), but since the qSDV is applied to CO2 and O2 it would be good to see where the differences mentioned (0.004) are coming from, from CO2 or O2.

To address this comment we have added another figure (Figure 7) which shows XAIR calculated using the column of O2 retrieved with the Voigt and the qSDV. Ideally XAIR should be 1 since the column of O2 is being used as a proxy for the dry column of air when calculating XCO2. However as shown in Figure 7 it is not. When using a Voigt lie shape to retrieve the O2 column, XAIR is 2% lower than it should be (at the smallest SZA) and has an airmass dependence that decreases as SZA increases (so the retrieved O2 column increases as SZA increases). By using the qSDV to retrieve the O2 column, less O2 is retrieved which results in the O2 column decreasing by 0.8% at the smallest SZA and up to 1.8% at the highest SZA as shown in Figure 6. Thus when the qSDV is used to retrieve O2 XAIR is closer to 1.The airmass dependence of XAIR changes when O2 is retrieved with the qSDV, causing XAIR to now increase as the airmass increases. The airmass dependence of the O2 column is thus similar to the airmass dependence of the CO2 column, so when calculating XCO2 with the column of O2 retrieved using the qSDV, the airmass dependence of XCO2 is minimized as shown in the new Figure 8.

We have added the following on lines 254-261: "Figure 7 shows XAIR from Park Falls on June 18, 2013. XAIR is the column of air (determined using surface pressure recorded at the site) divided by the column of O2 retrieved from the spectra and multiplied by 0.2095, which is the dry air mole fraction of O2 in Earth's atmosphere. Ideally XAIR should be 1 but when using O2 retrieved with a Voigt line shape (red points) it is closer to 0.98 near noon (small SZA) and lower near the start and end of the day (large SZA). When using O2 retrieved with the qSDV, XAIR is closer to 0.988 near noon and a bit higher near the start and end of the day. This means the O2 column, retrieved with the qSDV, decreases as a function of SZA, while previously the column increased as a function of SZA when the Voigt line shape is used."

We address the comment about the 0.004 change in comment 4.

Comment 4 - I found it a bit disappointing that the airmass dependance is now + 0.004 instead of - 0.013. This is a large reduction, but the results show that still something is wrong in the measurements/retrieval. The authors might discuss this in more detail. See above at 3.

The positive bias that now exists with the new spectroscopy is because the retrieved columns of $CO_2$ have increased when retrieved using the qSDV with line mixing while the retrieved columns of $O_2$ have decreased with the qSDV. This combination of an increase in the $CO_2$ column with a decrease in the $O_2$ column results in an increase in $XCO_2$. The decrease in retrieved $O_2$ column is good as noted in comment 3 but still needs to decrease further to match the column of dry air calculated from surface pressure measured at the TCCON stations. So if the retrieved $O_2$ column decreased further the positive bias between TCCON and the aircraft measurements would increase. This means that the retrieved columns of $CO_2$ are too high but for now compensate for the fact that the retrieved $O_2$ columns are still larger than they should be.

We have added the following to discuss this point on lines 319-324: "This increase in the slope can be explained by an increase in the retrieved column of $CO_2$ when using the qSDV with line mixing as shown in Mendonca et al. (2016) as well as combined with a decrease in the retrieved $O_2$ column due to using the qSDV. As discussed previously (section 5) the decrease in the retrieved $O_2$ column is an improvement but the expected column of $O_2$ is still approximately 1.2% higher (at the smallest SZA) than it should be. Therefore, the retrieved column of $CO_2$ is higher than it should be, and the slope would be greater if the retrieved column of $O_2$ was 1.2% lower."

Comment 5 - For me the fact that the airmass dependence is nearly gone when applying qSDV (Figure 8) very important. This should be more highlighted as main result.

**AMTD**

Figure 8 c and d look very similar. For me an airmass correction is not necessary, or is this a mistake in the panels?

Figures 8c and 8d are now Figures 10c and 10d. Figures 10c and 10d do look very similar but there is still an airmass dependence given by the fact that the correction term is not 0. It is now -0.0012, which is smaller than with a Voigt, but we still need to apply the correction to the data to account for this small airmass dependence. Since the airmass dependence has been significantly decreased Figures 10c and d look similar but are not the same.

Comment 6 - May be a Figure showing XCO2/O2 as a function of SZA for i) XCO2/O2, ii) XCO2(sQDV)/O2 ii) XCO2/O2(sQDV), iii) XCO2 (qSDV)/O2(qSDV) would be interesting to see where the improvement is coming from. For me a few other Figures of 1-5 could be deleted or put in the supplement.

We have added Figure 8 to show how the changes to the retrieved CO2 or O2 affect their airmass dependence. Figure 8 shows that the improvement of the retrieved column of O2 has a greater impact than the improvements made to the retrieval of CO2. However, the improvement made to the retrieval of CO2 are more critical at large SZA because it makes the airmass dependence of the column of CO2 for large SZA consistent with that for small SZA, allowing the SZA restriction on measurements at large SZA to be removed.

We have added the following lines 265-278: "Figure 8 is XCO2 calculated for four different combinations pertaining to the two CO2 column retrievals and the O2 column retrievals. The CO2 columns were retrieved with either a Voigt line shape (the standard GGG2014 approach) or the qSDV with line mixing as done in Mendonca et al. (2016) while the O2 columns were retrieved with either a Voigt (the standard GGG2014 approach) or the new qSDV approach developed here. Figure 8 shows a spurious symmetric component to XCO2 when the total column of O2 is retrieved with the Voigt line shape, regardless of line shape used to retrieve CO2. When the qSDV is used to retrieve total columns of O2, the symmetric component of XCO2 is diminished regardless of line shape used to retrieve CO2. This is because the airmass dependence of the column of O2 retrieved using the qSDV is more consistent with the airmass dependence of the column of CO2 (for both line shapes used to retrieve CO2). Mendonca et al. (2016) showed that using the qSDV with line mixing results in better fits to the CO2 windows and impacts the airmass dependence of the retrieved column of CO2. When using a Voigt line shape the retrieved column amount of CO2 decreases as airmass increases until the airmass is large (SZA of about 82o) at which point the retrieved column of CO2 increases as the airmass increases, changing the shape of the airmass dependence of the CO2 column. When the qSDV with line mixing is used, the retrieved column of CO2 decreases as a function of airmass (up until the sun is above the horizon)."

It is important that the Figures 1-5 remain since they show that the retrieved spectroscopic parameters have a dependence on quantum number m which has been shown to be the case in other studies of the discrete lines of the O2 1.27 $\mu$m electronic transitions.

Comment 7 – The main part of the paper deals with the speed-dependent Voigt line shape. I would suggest to include this in the title, may by: Improving the Retrieval of XCO2 from Total Carbon Column Network Solar Spectra by inclusion of the speed-dependent Voigt line shape.

Changed the title to: "Using a Speed-Dependent Voigt Line Shape to Retrieve O2 from Total Carbon Column Observing Network Solar Spectra to Improve Measurements of XCO2"

Comment 8 - In the conclusions the authors write: Using cavity ring-down spectra measured in the lab, we have shown that the Voigt line shape is insufficient to 290 model the line shape of O2 for the 1.27 $\mu$m band, ... As far as I see, the improvement might also results because the qSDV is applied also the CO2.

The improvement made to the retrieval of O2 has had an impact on the airmass dependence of XCO2 at all SZA while the improvements to the retrieval of CO2 has mainly impacted the airmass dependence at high SZA. As shown in Figure 8 using the O2 columns retrieved with the qSDV decreased the airmass dependence of XCO2 regardless of the line shape used to retrieve the CO2 columns. However, improvements made to the CO2 retrievals results in better measurements of XCO2 at high SZA. See comment 6 for discussion on this.

Please also note the supplement to this comment:
https://www.atmos-meas-tech-discuss.net/amt-2018-62/amt-2018-62-AC2-supplement.pdf

**Supplement:**

[revised manuscript text omitted]

**O$_2$ columns retrieved using a Voigt (red) and qSDV (green) line shapes.**

[Figure]

**Figure 8: XCO₂ calculated from the CO₂ and O₂ columns retrieved from Park Falls spectra recorded on June 18, 2013. The CO₂ columns were retrievied using either the Voigt line shape or the qSDV with line mixing, while the O₂ columns were retrieved using either the Voigt or qSDV line shapes. XCO₂ was calculated in four ways: 1) Both CO₂ and O₂ columns retrieved using the Voigt line shape (red), 2) CO₂ columns retrieved with the Voigt and O₂ columns retrieved with the qSDV (green), 3) CO₂ columns retrieved with the qSDV and line mixing and O₂ columns retrieved with the Voigt (cyan), and 4) CO₂ columns retrieved with the qSDV and line mixing and O₂ columns retrieved with the qSDV (blue).**

[Figure]

**Figure 9: The average airmass-dependent correction factor for XCO₂ derived from a year of spectra**
**measured at Darwin, Lamont, and Park Falls for different seasons. The dashed lines with stars are the α for**
**XCO₂ retrieved using a Voigt line shape for both CO₂ and O₂ columns. The solid lines with squares are from**
**XCO₂ retrieved using the qSDV for both CO₂ and O₂ columns.**

[Figure]

**Figure 108: (a) XCO₂ Voigt anomaly for a year of measurements from the four TCCON sites. The XCO₂**
**anomaly is the difference between each XCO₂ value and the daily median XCO₂. (b) The XCO₂ Voigt**
**anomaly after the airmass dependence correction is applied to the XCO₂ Voigt data. (c) XCO₂ qSDV**
**anomaly. (d) XCO₂ qSDV anomaly after correction for the airmass dependence.**

[Figure]

**Figure 119: (a) Correlation between TCCON and aircraft XCO₂ Voigt measurements for 13 TCCON sites. Each aircraft type is indicated by a different symbol given by the legend in the top left corner. Each site is represented by a different colour given by the legend in the bottom right corner. The grey line indicates the one-to-one line and the dashed line is the line of best fit for the data. The slope of the line of best fit as well as the error on the slope are given in the plot. (b) the same as (a) but for XCO₂ qSDV.**

[Figure]

**Figure 12̶1̶0̶: (a) to (d) XCO₂ plotted as a function of day of the year for Eureka (2014), Park Falls (2013),**
**Lamont (2010), and Darwin (2006) respectively. The red boxes are XCO₂ calculated from using a Voigt line**
**shape in the retrieval and the blue boxes are from using the qSDV. (e) to (h) the difference between XCO₂**
**Voigt and XCO₂ qSDV.**

[Figure]

**Figure 1311: (a) XCO₂ from Park Falls retrieved from spectra recorded on June 18, 2013. Plotted is XCO₂**
**retrieved: (1) with a Voigt line shape and corrected for the airmass dependence (red squares), (2) with the**
**qSDV (cyan circles), and (3) with the qSDV and corrected for the airmass dependence (blue squares). (b) the**
**difference between the Voigt XCO₂ corrected and the qSDV XCO₂ (cyan circles), and the difference between**
**the Voigt XCO₂ corrected and the qSDV XCO₂ corrected (blue squares).**

---

## Author Response (AR1)

Referee comments are given in *Blue*.

Response to comments are given in Black.

In the manuscript highlighted text is the added text and  text is deleted text.

**Response to Referee 1**

We thank the reviewer for the comments on our manuscript. Please see below for our
responses.

*Comment 1 – In lines 55-62 authors talk about different works on the A-band and mentioning*
*Galatry retrievals. However, for some reason the work of Drouin et al (JQSRT 2016) was not*
*mentioned although it also employed the SDV profile.*
We acknowledge that the work by Drouin et al. (2017) uses a SDV profile when fitting the A-
band and have included it in the introduction of when discussing the line shape work done with
the $O_2$ A-band.
We have added the following on lines 80-82: "When fitting cavity ring-down spectra of the $O_2$
A-band, Drouin et al. (2017) found it necessary to use a speed-dependence Voigt line shape,
which takes into account different speeds at the time of collision (Shannon et al., 1986), with
line mixing to properly fit the discrete spectral lines of the $O_2$ A-band."
*Comment 2 – The authors may want to mention that the HITRAN2016 parameters are very*
*similar to those in HITRAN2012 in this particular band and the only change are improved line*
*positions from Yu et J. Chem. Phys. 141 (2014) 174302. doi:10.1063/1.4900510.*
We acknowledge that this should be included in the introductory section about the discrete $O_2$
1.27 µm band since it is the latest version of the spectroscopic parameters used for this band.
We have added the following on lines 64-66: "Spectroscopic parameters for the discrete
spectral lines of the $O_2$ 1.27 µm band  from HITRAN 2016 (Gordon et al., 2017) are very similar
to HITRAN 2012 except that HITRAN2016 includes improved line positions reported by Yu et al.
(2014)."
*Comment 3 – It is interesting that the authors do mention the line-mixing with respect to the*
*Aband and CO2 bands but did not say about this effect in 1.27 micron band that they*
*investigated. It is also not mentioned as potential source of remaining residuals in lines*
*293-303. It would be interesting to see some discussion about this.*

We have added a discussion about line mixing and how it impacts some of the retrieved
spectroscopic parameters as well as the remaining residuals seen when fitting the lab spectra.

We have added the following on lines 359-367: "This can be explained by the fact that line mixing,
which is shown to be important for the $O_2$ A-band, was not considered when fitting the cavity-ringdown
spectra. Neglecting line mixing usually produces an asymmetric residual in the discrete lines as well as a
broad residual feature associated with the fact that collisions are transferring intensity from one part of
the spectrum to another. By fitting a set of Legendre polynomials for CIA we could simultaneously be
fitting the broader band feature associated with line mixing while the retrieved pressure shifts, and
speed-dependent pressure shifts could be compensating for the asymmetric structure one would see in
the discrete lines when neglecting line mixing. The remaining structure, as seen in Figure 1c, could be
due to neglecting line mixing especially in the Q-branch where the spacing between spectral lines is
small (in comparison to the P and R branches) and line mixing is most likely prevalent."

*Comment 4 – Talking about the sources of the residuals and its potential relation to Dicke*
*narrowing it would be interesting what authors think about conclusions of the Torun group*
*(Domyslawska et al papers in JQSRT 2014-2016), that for the electronic transitions of*
*O2 speed-dependence should have much larger effect than Dicke narrowing.*

To address this comment we have added the following discussion on lines 371-383:
"Domysławska et al. (2016) recommend using the qSDV to model the line shape of $O_2$ based on
multiple line shape studies of the $O_2$ B-band. In these studies, a multi-spectrum fit to low
pressure (0.27-5.87 kPa) cavity-ring down spectra was preformed testing multiple line shapes
that took speed-dependence and Dicke narrowing into account both separately and
simultaneously. They found that the line shapes that only used Dicke narrowing were not good
enough to model the line shape of the $O_2$ B-band lines, but a line shape that included either
speed-dependence or both speed-dependence and Dicke narrowing produced similar quality
fits, ultimately concluding that speed-dependence has a larger effect than Dicke narrowing. It
was noted in the study by Wójtewicz et al., (2014) that both Dicke narrowing and speed-
dependent effects might simultaneously play an important role in modeling the line shape of
the $O_2$ B-band lines. However, the speed-dependent and Dicke narrowing parameters are highly
correlated at low pressures. To reduce the correlation requires either a multi-spectrum fit of
spectra at low pressures with high enough signal to nosie ratio or spectra that cover a wide
range of pressure (Wójtewicz et al., 2014). So, by combining the high-pressure spectra used in
this study with low pressure spectra in a multipspectrum fit both the speed-dependence and
Dicke narrowing parameters could be retrieved."

*Comment 5 – Spectral shifts in the 1.27 micron band had always been very hard to measure.*
*See for instance discussion in Hill et al, J. Mol. Spectrosc. 221 (2003) 286–287.*
*doi:10.1016/S0022-2852(03)00227-3 and Newman, et al, J. Chem. Phys. 110 (1999)*
*10749. doi:10.1063/1.479018.*

*The authors may want to mention this. Continuing the topic of shifts it is well known that*
*while the widths in P and R branches for same rotational quanta should be very similar*

the shifts should be assymetric. Therefore I would suggest to plot these separately
or using running number m, where m=-J for P lines and J+1 for R. The authors may
also want to use the upper state rotational quanta because they are not split into spin
components.

We have added the following on lines 356-359, to address this comment: "Accurate
measurements of the pressure shifts in the 1.27 µm band have been hard to obtain as shown in
Newman et al. (1999) and Hill et al., (2003). While the retrieved pressure shifts show a
dependence on quantum number m (Figure 3c) as one would expect, this dependence is not as
strong as the m dependence of the Lorentz widths (Figure 3b)."

We have also replotted Figures 3 and 4 to show the retrieved parameters as a function of m.

*Comment 6 - Does one need to account for airglow when analysing the 1.27 TCCON spectra?*
*See Sun et al (https://doi.org/10.1029/2018GL077823) for instance, regarding significance*
*of airglow in oxygen's 1.27 micron band at the top of atmosphere.*

Since TCCON spectra are recorded by viewing the sun directly, airglow emission is negligible
since the signal from the sun is much more intense than airglow.

We have added the following on lines 220-221: "Airglow is not considered when fitting the 1.27
µm band since the spectrometer views the sun directly, and airglow is overwhelmed by such a
bright source."

**Response to Referee 2**

We thank the reviewer for the comments on our manuscript. Please see below for our response
to the comments.

*Comment 1 - The paper is very theoretical. Line 92 starts with: to take speed dependence into*
*account … Here it would be nice to explain what is meant by speed dependence. It should be*
*mentioned that the assumed basis for the Lorentz portion of the Voigt profile is, that for all*
*collisions between the molecules the statistical average velocity is taken. However, in reality*
*this is not true, the molecules have a distribution of speeds, which requires the qSDV.*
To address this we have added the following:
Lines 104-105: "The Voigt line shape is the convolution of the Lorentz and the Gaussian profiles,
which model pressure and doppler broadening of the spectral line respectively."
Lines 112-113: "The Voigt line shape assumes that pressure broadening is accurately
represented by a Lorentz profile calculated for the stastical average velocity at the time of
collission."

Lines 115-118: "The speed-dependent Voigt line shape refines the pressure broadening
component of the Voigt by calculating multiple Lorentz profiles for different speeds at the time
of collision. The final contribution from pressure broadening to the speed-dependent Voigt is
the weighted sum of Lorentz profiles (weighted by the Maxwell-Boltzmann speed-distribution)
calculated for different speeds at the time of collision."

*Comment 2* - The same holds for the Dicke narrowing, mentioned in line 59. What is the Dicke
narrowing? It should be mentioned that when the mean free path of an atom is much smaller
than the wavelength of the radiative transition, the atom changes velocity and direction many
times during the emission or absorption of a photon. This causes an averaging over different
Doppler states and results in an atomic linewidth that is narrower than the Doppler width (I
have taken this from Wikipedia).

To address this comment we have added the following:

Lines 71-74: "Dicke narrowing occurs when the motion of the molecule is diffusive due to
collisions changing the velocity and direction of the molecule during the time that it is excited.
This diffusive motion is taken into account by averaging over many different Doppler states
resulting in a line width that is narrower than the Doppler width (Dicke, 1953)."

*Comment 3* - The O2 concentration in the atmosphere is very stable and well known. I would be
interested to see the difference between the known O2 concentration and the O2 from the
TCCON spectra as a function of the SZA. These results are somehow hidden in the paper (Figure
6), but since the qSDV is applied to CO2 and O2 it would be good to see where the differences
mentioned (0.004) are coming from, from CO2 or O2.

To address this comment we have added another figure (Figure 7) which shows XAIR calculated
using the column of $O_2$ retrieved with the Voigt and the qSDV. Ideally XAIR should be 1 since the
column of $O_2$ is being used as a proxy for the dry column of air when calculating $XCO_2$. However
as shown in Figure 7 it is not. When using a Voigt lie shape to retrieve the $O_2$ column, XAIR is 2%
lower than it should be (at the smallest SZA) and has an airmass dependence that decreases as
SZA increases (so the retrieved $O_2$ column increases as SZA increases). By using the qSDV to
retrieve the $O_2$ column, less $O_2$ is retrieved which results in the $O_2$ column decreasing by 0.8% at
the smallest SZA and up to 1.8% at the highest SZA as shown in Figure 6. Thus when the qSDV is
used to retrieve $O_2$ XAIR is closer to 1.The airmass dependence of XAIR changes when $O_2$ is
retrieved with the qSDV, causing XAIR to now increase as the airmass increases. The airmass
dependence of the $O_2$ column is thus similar to the airmass dependence of the $CO_2$ column, so
when calculating $XCO_2$ with the column of $O_2$ retrieved using the qSDV, the airmass dependence
of $XCO_2$ is minimized as shown in the new Figure 8.

We have added the following on lines 254-261: "Figure 7 shows XAIR from Park Falls on June
18, 2013. XAIR is the column of air (determined using surface pressure recorded at the site)
divided by the column of $O_2$ retrieved from the spectra and multiplied by 0.2095, which is the
dry air mole fraction of $O_2$ in Earth's atmosphere. Ideally XAIR should be 1 but when using $O_2$

retrieved with a Voigt line shape (red points) it is closer to 0.98 near noon (small SZA) and lower
near the start and end of the day (large SZA). When using $O_2$ retrieved with the qSDV, XAIR is
closer to 0.988 near noon and a bit higher near the start and end of the day. This means the $O_2$
column, retrieved with the qSDV, decreases as a function of SZA, while previously the column
increased as a function of SZA when the Voigt line shape is used."

We address the comment about the 0.004 change in comment 4.

*Comment 4* - I found it a bit disappointing that the airmass dependance is now + 0.004 instead
of - 0.013. This is a large reduction, but the results show that still something is wrong in the
measurements/retrieval. The authors might discuss this in more detail. See above at 3.

The positive bias that now exists with the new spectroscopy is because the retrieved columns
of $CO_2$ have increased when retrieved using the qSDV with line mixing while the retrieved
columns of $O_2$ have decreased with the qSDV. This combination of an increase in the $CO_2$
column with a decrease in the $O_2$ column results in an increase in $XCO_2$. The decrease in
retrieved $O_2$ column is good as noted in comment 3 but still needs to decrease further to match
the column of dry air calculated from surface pressure measured at the TCCON stations.  So if
the retrieved $O_2$ column decreased further the positive bias between TCCON and the aircraft
measurements would increase. This means that the retrieved columns of $CO_2$ are too high but
for now compensate for the fact that the retrieved $O_2$ columns are still larger than they should
be.

We have added the following to discuss this point on lines 319-324: "This increase in the slope
can be explained by an increase in the retrieved column of $CO_2$ when using the qSDV with line
mixing as shown in Mendonca et al. (2016) as well as combined with a decrease in the retrieved
$O_2$ column due to using the qSDV. As discussed previously (section 5) the decrease in the
retrieved $O_2$ column is an improvement but the expected column of $O_2$ is still approximately
1.2% higher (at the smallest SZA) than it should be. Therefore, the retrieved column of $CO_2$ is
higher than it should be, and the slope would be greater if the retrieved column of $O_2$ was 1.2%
lower."

*Comment 5* - For me the fact that the airmass dependence is nearly gone when applying qSDV
(Figure 8) very important. This should be more highlighted as main result. Figure 8 c and d look
very similar. For me an airmass correction is not necessary, or is this a mistake in the panels?

Figures 8c and 8d are now Figures 10c and 10d. Figures 10c and 10d do look very similar but
there is still an airmass dependence given by the fact that the correction term is not 0. It is now
-0.0012, which is smaller than with a Voigt, but we still need to apply the correction to the data
to account for this small airmass dependence. Since the airmass dependence has been
significantly decreased Figures 10c and d look similar but are not the same.

*Comment 6* - May be a Figure showing XCO2/O2 as a function of SZA for i) XCO2/O2, ii)
XCO2(sQDV)/O2 ii) XCO2/O2(sQDV), iii) XCO2 (qSDV)/O2(qSDV) would be interesting to see where the improvement is coming from. For me a few other Figures of 1-5 could be deleted or
put in the supplement.

We have added Figure 8 to show how the changes to the retrieved $CO_2$ or $O_2$ affect their
airmass dependence. Figure 8 shows that the improvement of the retrieved column of $O_2$ has a
greater impact than the improvements made to the retrieval of $CO_2$. However, the
improvement made to the retrieval of $CO_2$ are more critical at large SZA because it makes the
airmass dependence of the column of $CO_2$ for large SZA consistent with that for small SZA,
allowing the SZA restriction on measurements at large SZA to be removed.

We have added the following lines 265-278: "Figure 8 is $XCO_2$ calculated for four different
combinations pertaining to the two $CO_2$ column retrievals and the $O_2$ column retrievals. The
$CO_2$ columns were retrieved with either a Voigt line shape (the standard GGG2014 approach) or
the qSDV with line mixing as done in Mendonca et al. (2016) while the $O_2$ columns were
retrieved with either a Voigt (the standard GGG2014 approach) or the new qSDV approach
developed here. Figure 8 shows a spurious symmetric component to $XCO_2$ when the total
column of $O_2$ is retrieved with the Voigt line shape, regardless of line shape used to retrieve
$CO_2$. When the qSDV is used to retrieve total columns of $O_2$, the symmetric component of $XCO_2$
is diminished regardless of line shape used to retrieve $CO_2$. This is because the airmass
dependence of the column of $O_2$ retrieved using the qSDV is more consistent with the airmass
dependence of the column of $CO_2$ (for both line shapes used to retrieve $CO_2$). Mendonca et al.
(2016) showed that using the qSDV with line mixing results in better fits to the $CO_2$ windows
and impacts the airmass dependence of the retrieved column of $CO_2$. When using a Voigt line
shape the retrieved column amount of $CO_2$ decreases as airmass increases until the airmass is
large (SZA of about 82°) at which point the retrieved column of $CO_2$ increases as the airmass
increases, changing the shape of the airmass dependence of the $CO_2$ column. When the qSDV
with line mixing is used, the retrieved column of $CO_2$ decreases as a function of airmass (up
until the sun is above the horizon)."

It is important that the Figures 1-5 remain since they show that the retrieved spectroscopic
parameters have a dependence on quantum number m which has been shown to be the case in
other studies of the discrete lines of the $O_2$ 1.27 μm electronic transitions.

*Comment 7 –* The main part of the paper deals with the speed-dependent Voigt line shape. I
would suggest to include this in the title, may by: Improving the Retrieval of XCO2 from Total
Carbon Column Network Solar Spectra by inclusion of the speed-dependent Voigt line shape.

Changed the title to: "Using a Speed-Dependent Voigt Line Shape to Retrieve $O_2$ from Total
Carbon Column Observing Network Solar Spectra to Improve Measurements of $XCO_2$"

*Comment 8 -* In the conclusions the authors write: Using cavity ring-down spectra measured in
the lab, we have shown that the Voigt line shape is insufficient to 290 model the line shape of
O2 for the 1.27 μm band, ... As far as I see, the improvement might also results because the
qSDV is applied also the CO2.

The improvement made to the retrieval of $O_2$ has had an impact on the airmass dependence of
$XCO_2$ at all SZA while the improvements to the retrieval of $CO_2$ has mainly impacted the airmass
dependence at high SZA. As shown in Figure 8 using the $O_2$ columns retrieved with the qSDV
decreased the airmass dependence of $XCO_2$ regardless of the line shape used to retrieve the
$CO_2$ columns. However, improvements made to the $CO_2$ retrievals results in better
measurements of $XCO_2$ at high SZA. See comment 6 for discussion on this.

**Additional Changes to the Manuscript**

In the introduction, added a reference to Wallace and Livingston (1990) since they were the
first to measure $O_2$ from this band.

Added references for the TCCON data versions and GGG2014 spectral line list versions.

Changed Figures 1b, 1c, 2b, and 2c as the scale on the y-axis was wrong since the units for the
residuals were not consistent with the units used in Figures 1a and 2a.

Corrected x-axis of Figure 13.

Some minor typographical and phrasing corrections were made.

[revised manuscript text omitted]
 18, 2013. XAIR is calculated using O₂ columns retrieved using a Voigt (red) and qSDV (green) line shapes. The top x-axis is the SZA that corresponds to the hour on the bottom x-axis.**

[Figure]

**Figure 8: XCO₂ calculated from the CO₂ and O₂ columns retrieved from Park Falls spectra recorded on June 18, 2013. The CO₂ columns were retrievied using either the Voigt line shape or the qSDV with line mixing, while the O₂ columns were retrieved using either the Voigt or qSDV line shapes. XCO₂ was calculated in four ways: 1) Both CO₂ and O₂ columns retrieved using the Voigt line shape (red), 2) CO₂ columns retrieved with the Voigt and O₂ columns retrieved with the qSDV (green), 3) CO₂ columns retrieved with the qSDV and line mixing and O₂ columns retrieved with the Voigt (cyan), and 4) CO₂ columns retrieved with the qSDV and line mixing and O₂ columns retrieved with the qSDV (blue). The top x-axis is the SZA that corresponds to the hour on the bottom x-axis.**

[Figure]

**Figure 9̶7̶: The average airmass-dependent correction factor for XCO₂ derived from a year of spectra**
**measured at Darwin, Lamont, and Park Falls for different seasons. The dashed lines with stars are the α for**
**XCO₂ retrieved using a Voigt line shape for both CO₂ and O₂ columns. The solid lines with squares are from**
**XCO₂ retrieved using the qSDV for both CO₂ and O₂ columns.**

[Figure]

**Figure 108: (a) XCO$_2$ Voigt anomaly for a year of measurements from the four TCCON sites. The XCO$_2$**
**anomaly is the difference between each XCO$_2$ value and the daily median XCO$_2$. (b) The XCO$_2$ Voigt**
**anomaly after the airmass dependence correction is applied to the XCO$_2$ Voigt data. (c) XCO$_2$ qSDV**
**anomaly. (d) XCO$_2$ qSDV anomaly after correction for the airmass dependence.**

[Figure]

**Figure 11: (a) Correlation between TCCON and aircraft XCO₂ Voigt measurements for 13 TCCON sites. Each aircraft type is indicated by a different symbol given by the legend in the top left corner. Each site is represented by a different colour given by the legend in the bottom right corner. The grey line indicates the one-to-one line and the dashed line is the line of best fit for the data. The slope of the line of best fit as well as the error on the slope are given in the plot. (b) the same as (a) but for XCO₂ qSDV.**

[Figure]

**Figure 12̶1̶0̶: (a) to (d) XCO₂ plotted as a function of day of the year for Eureka (2014), Park Falls (2013),**
**Lamont (2010), and Darwin (2006) respectively. The mostly-hidden red boxes are XCO₂ calculated from**
**using a Voigt line shape in the retrieval and the blue boxes are from using the qSDV. (e) to (h) the difference**
**between XCO₂ Voigt and XCO₂ qSDV.**

[Figure]

**Figure 13̶1̶1̶: (a) XCO₂ from Park Falls retrieved from spectra recorded on June 18, 2013. Plotted is XCO₂**
**retrieved: (1) with a Voigt line shape and corrected for the airmass dependence (red squares), (2) with the**
**qSDV (cyan circles), and (3) with the qSDV and corrected for the airmass dependence (blue squares). (b) the**
**difference between the Voigt XCO₂ corrected and the qSDV XCO₂ (cyan circles), and the difference between**
**the Voigt XCO₂ corrected and the qSDV XCO₂ corrected (blue squares). The top x-axis is the SZA that**
**corresponds to the hour on the bottom x-axis.**

---

## Author Response (AR2)

Hi Frank

Thanks for the comments they were helpful in improving the manuscript.

Please see my response to the comments below. When looking for the line number used in the manuscript please refer to the non-marked up version.

Comment 1:

The first part of section 5.2 and Fig 11 heavily rely on data collected the framework of various aircraft campaigns. I would suggest to include campaign-specific citations here, e.g. for IMECC

Messerschmidt, et al.: Calibration of TCCON column-averaged CO2: the first aircraft campaign over European TCCON sites, Atmospheric Chemistry and Physics, 11(21), 10765-10777, doi:10.5194/acp-11-10765-2011.

Since we have included citations for the TCCON data sets we acknowledge we should include ciations for the aircraft campaigns.

To address this, we have added the following citations on line 306:

Deutscher et al., 2010; Lin et al., 2006; Messerschmidt et al., 2010; Singh et al., 2006; Wofsy, 2011

Comment 2:

Figure 7 included upon request of referee 2 is by far not as informative as it could be. Would you please replace it by separate panels showing XAIR as fct of SZA for the complete datasets, using old and the new linelists? This would demonstrate in a very convining manner that XAIR is improved (flatter) with the new linelist. Ideally, show separate panels (adding some vertical offset between old and new results for clarity if needed) for each site to reveal any systematic differences between sites (which ideally should not exist).

We have changed Figure 7 to show XAIR plotted as a function of SZA for the entire data set.

The text on lines 244-254 was changed to the following:

"Figure 7 shows XAIR for the entire data set plotted as a funtion of SZA. XAIR is the column of air (determined using surface pressure recorded at the site) divided by the column of $O_2$ retrieved from the spectra and multiplied by 0.2095, which is the dry air mole fraction of $O_2$ in Earth's atmosphere. Ideally XAIR should be 1 but when using $O_2$ retrieved with a Voigt line shape (Figure 7a) to calculate XAIR the average XAIR value for the entire data set is 0.977. Using $O_2$ retrieved with the qSDV, to calculate XAIR, the average value is 0.986 which is closer to the expected value of 1. However, XAIR has a dependence on SZA regardless of line shape used. Figure 7a shows that XAIR decreases as a function of SZA (evident at SZA$> 75°$) which means that the retrieved column of $O_2$ increases as a function of SZA. Figure 7b shows that XAIR increases as a function of SZA (evident at SZA$> 70°$), which means that the retrieved column of $O_2$ decreases as a function of SZA. Therefore retriving total columns of $O_2$ with the qSDV changes the airmass dependendnce of the $O_2$ column which in turn will impact the airmass dependence of $XCO_2$."

The purpose of Figure 7 is to show that (1) XAIR is now closer to the expected value of 1 which is an improvement in the retrieval and (2) that regardless of line shape used the retrieved total column of $O_2$ has an airmass dependence regardless of spectral. Using the qSDV does not make XAIR (or rather the retrieved total column of $O_2$) flatter but rather changes the airmass dependence of $O_2$. In section 5.1, we investigate how $O_2$ retrieved with the qSDV impacts the airmass dependence of $XCO_2$ which is shown that it decreases it. Since the total column of $CO_2$ has an airmass dependence (regardless of spectral line shape used to retrieve it) retrieving a total column of $O_2$ that was flat as a function of SZA would lead to $XCO_2$ that would still have an airmass dependence because the total column of $CO_2$ has an airmass dependence. Figure 8 is showing that the airmass dependence of $O_2$ retrieved with the qSDV is similar to the airmass dependence of $CO_2$ so when calculating $XCO_2$ the airmass dependence of both the $CO_2$ and $O_2$ columns almost cancels each other out.

The following text was added on lines 392-397 in the discussion and conclusions section to state that XAIR is now closer to 1 but and airmass dependence of the retrieved $O_2$ column still remains:

"XAIR calculated with the column of $O_2$ retrieved with the qSDV is now closer to the expected value of 1 but XAIR still has an airmass dependence which result of the retrieved total column of $O_2$ decreasing as a function of SZA at large SZA. This remaining airmass dependence could be due to neglecting affects such as Dicke narrowing and line mixing in the absorption coefficient calculations, as well as assuming a perfect instrument line shape in the retrieval algorithm. However, retrieving $O_2$ with the qSDV significantly decreases the airmass dependence of $XCO_2$."

Comment 3:

Do you think that the retrieved values for the speed dependent shift parameters are significant? The reported error bars are large for at least a subset of lines. Do you recommend for atmospheric work to use the value as reported, or to apply a smooth interpolation in m or to omit the parameters for all lines? Perhaps you could add a short comment on this point in the paper (or I overlooked...)?

To address this comment, we have added the following on line 359-369:

"The large error bars for the measured pressure shifts and speed-dependent pressure shifts as well as a deviation from a smooth m dependence of these parameters could be due to neglecting line mixing when fitting the lab spectra. Figure 3c and 3d show that the spectral lines that have large error bars and deviate from an expected m dependence belong mainly to the Q-branch spectral lines (which are mostly likely impacted by line mixing). To achieve the results obtained in this study it is best to use the parameters as is instead of trying to apply an interpolation, that depends on m, or even omitting them unless one test's these changes on atmospheric spectra that cover different range of conditions (i.e. seasons, dry/wet, SZA, geographical locations). It is evident that the parameters might be compensating for affects (such as line mixing) that were not included when fitting the lab spectra and changing these parameters (or omitting them) could lead to degradation in the quality of the spectral fits of solar spectra and change the airmass dependence of the retrieved column of $O_2$ which would impact the airmass dependence of $XCO_2$."

The conclusion "measurements made at SZA > 82 deg no longer have to been discarded" is a bit vague. Do you feel that the SZA range for TCCON should be extended to a new, higher limit (e.g. 86 deg), as supported by datasets from several sites (Fig. 10), or even accept the whole range of SZAs (then only Lamont data - becoming increasingly noisy - remain as supporting evidence)?

Comment 4:

To address this comment, we have added the following on line 398:

"We recommend using the full range of SZA which would result in more $XCO_2$ measurement available from all TCCON sites."

Comment 5:

There are a few typos in the new manuscript:

line 12: spectrometers that are part of
line 32: below
line 140: are two blanks before "Abrarov"?
line 214:red line in my printout after "2014a)"
line 338: missing blank before "(Hartmann"

These typos as well as other formatting issues have been fixed.

Thanks,

Joseph

[revised manuscript text omitted]